# Regulation of the endosomal SNX27-retromer by OTULIN

Aurelia Stangl[1,9], Paul R. Elliott[2,3,9], Adan Pinto-Fernandez[4], Sarah Bonham[4], Luke Harrison[5], Annalisa Schaub[1], Kerstin Kutzner[1], Kirstin Keusekotten[2], Paul T. Pfluger [5,6], Farid El Oualid [7], Benedikt M. Kessler [4], David Komander[2,8] & Daniel Krappmann [1]

OTULIN (OTU Deubiquitinase With Linear Linkage Specificity) specifically hydrolyzes methionine1 (Met1)-linked ubiquitin chains conjugated by LUBAC (linear ubiquitin chain assembly complex). Here we report on the mass spectrometric identification of the OTULIN interactor SNX27 (sorting nexin 27), an adaptor of the endosomal retromer complex responsible for protein recycling to the cell surface. The C-terminal PDZ-binding motif (PDZbm) in OTULIN associates with the cargo-binding site in the PDZ domain of SNX27. By solving the structure of the OTU domain in complex with the PDZ domain, we demonstrate that a second interface contributes to the selective, high affinity interaction of OTULIN and SNX27. SNX27 does not affect OTULIN catalytic activity, OTULIN-LUBAC binding or Met1-linked ubiquitin chain homeostasis. However, via association, OTULIN antagonizes SNX27-dependent cargo loading, binding of SNX27 to the VPS26A-retromer subunit and endosome-to-plasma membrane trafficking. Thus, we define an additional, non-catalytic function of OTULIN in the regulation of SNX27-retromer assembly and recycling to the cell surface.

[1] Research Unit Cellular Signal Integration, Institute of Molecular Toxicology and Pharmacology, Helmholtz Zentrum München, Ingolstaedter Landstrasse 1, 85764 Neuherberg, Germany. [2] Medical Research Council Laboratory of Molecular Biology, Francis Crick Avenue, Cambridge CB2 0QH, UK. [3] Department of Biochemistry, University of Oxford, South Parks Road, Oxford OX1 3QU, UK. [4] Target Discovery Institute, Nuffield Department of Medicine, University of Oxford, Roosevelt Drive, Oxford OX3 7FZ, UK. [5] Research Unit Neurobiology of Diabetes, Institute for Diabetes and Obesity, Helmholtz Zentrum München, German Centre for Diabetes Research (DZD), Ingolstaedter Landstrasse 1, 85764 Neuherberg, Germany. [6] TUM School of Medicine, Technische Universität München, 80333 Munich, Germany. [7] UbiQ Bio BV, Science Park 408, 1098 XH Amsterdam, The Netherlands. [8] Ubiquitin Signalling Division, The Walter and Eliza Hall Institute of Medical Research, 1 G Royale Parade, Parkville, Melbourne 3052, Australia. [9] These authors contributed equally: Aurelia Stangl, Paul R. Elliott. Correspondence and requests for materials should be addressed to P.R.E. (email: paul.elliott@bioch.ox.ac.uk) or to D.K. (email: dk@wehi.edu.au) or to D.K. (email: daniel.krappmann@helmholtz-muenchen.de)

The post-translational modifier ubiquitin controls most cellular processes[1,2]. Despite conjugation to substrate proteins, ubiquitin assembles covalent chains either through iso-peptide linkages utilizing the seven internal lysine (Lys) residues or through a peptide bond with N-terminal methionine1 (Met1). Conjugation of Met1-linked polyubiquitin (polyUb) is catalyzed by the linear ubiquitin chain assembly complex (LUBAC), consisting of HOIP/RNF31, HOIL-1/RBCK1, and SHARPIN[3,4]. Met1-linked polyUb is hydrolyzed by deubiquitinating enzymes (DUBs) such as OTULIN or CYLD[5–7].

OTULIN with its catalytic OTU (ovarian tumor) domain is highly specific in binding and hydrolyzing Met1-linked ubiquitin chains[5,8]. OTULIN directly binds to the N-terminal HOIP PUB (peptide:N-glycanase/UBA- or UBX-containing proteins) domain through a highly conserved PUB-interacting motif (PIM)[9,10]. Expression of OTULIN prevents auto-ubiquitination of LUBAC components at steady state and cellular accumulation of Met1-linked polyUb[5,9–11]. Furthermore, OTULIN restricts LUBAC activity and NF-κB signaling in response to TNFα and NOD2 stimulation[5,11]. The N-terminus of OTULIN also interacts with disheveled 2 (DVL2) and modulates in conjunction with LUBAC canonical WNT signaling[7,12]. Embryonal lethality was observed in mice lacking OTULIN, defective in ubiquitin-binding (OTULIN W96R) and devoid of catalytic activity (OTULIN C129A)[7,13,14]. Conditional deletion of OTULIN in myeloid cells leads to accumulation of Met1-linked ubiquitin chains, constitutive NF-κB activation and chronic inflammation as well as autoimmunity resembling the human ORAS (OTULIN—related auto-inflammatory syndrome) caused by hypomorphic OTULIN germline mutations[13,15]. Adult OTULIN C129A mutant mice exhibit auto-inflammation, which relies on aberrant induction of cell death[14]. Importantly, so far all functions of OTULIN have been attributed to its role in ubiquitin homeostasis by catalyzing deconjugation of Met1-linked ubiquitin chains and LUBAC binding.

SNX27 (sorting nexin 27) belongs to the class of retromer-associated sorting nexins localized on early endosomes[16]. Upon internalization of many plasma membrane proteins, SNX27-containing retromers are specifically responsible for the fast recycling of these cargos from the endosomes to the plasma membrane[16]. Through its PDZ (PSD95–Dlg1–ZO-1) domain, SNX27 recruits membrane cargos and other proteins via their C-terminal PDZ-binding motifs (PDZbms)[17,18]. In mice, loss of SNX27 causes growth defects leading to death within 4 weeks after birth[19]. Interestingly, the C-terminal amino acids Glu349-Thr350-Ser351-Leu352 (ETSL) in OTULIN correspond to a canonical class I PDZbm[16].

Through mass spectrometry approaches, we identified SNX27 as an interactor of OTULIN. Using structural, biophysical, and cellular analyses, we demonstrate that OTULIN binds with high affinity to two distinct surfaces in the SNX27 PDZ domain and thereby competes in a non-catalytic manner for cargo binding, retromer assembly, and endosome-to-membrane trafficking. Thus, despite its known function in regulating ubiquitin homeostasis, we define a new role for OTULIN in controlling SNX27-dependent endosomal sorting processes.

## Results

**Identification of SNX27 as an OTULIN interactor.** To identify potential OTULIN interactors, we took advantage of the biotin-labeled OTULIN activity-based probe (ABP) (bio-UbG76Dha-Ub), which couples with high selectivity to active OTULIN in cell extracts[20]. Extracts from Jurkat T cells were incubated with OTULIN ABP and subjected to pull downs (OTULIN ABP-PD). ATP was depleted from all extracts to prevent the coupling of the

OTULIN ABP to E1 enzymes and the conjugation into ubiquitin chains[20]. Proteins enriched after OTULIN ABP-PD were identified by label-free quantitative liquid chromatography-tandem mass spectrometry (LFQ LC-MS/MS). A volcano plot comparing control (no probe) versus OTULIN ABP-PD showed that besides the OTULIN ABP (UBC: polyUb precursor) itself and OTULIN, 10 other proteins were significantly enriched in the OTULIN ABP-PD samples (Fig. 1a). Among these candidates were the DUBs UCHL3 and USP5 that were previously identified to bind non-covalently to the linear diubiquitin probe[20]. Many enriched proteins have been associated with ubiquitin regulation (HOIP, BRAP2, USP3, NEDD8, and VCP/p97) or contain ubiquitin-binding domains (HDAC6 and WRNIP1). Only SNX27 has not been directly connected to the ubiquitin system. Importantly, besides OTULIN, only HOIP and SNX27 were significantly depleted from the ABP-bound fraction by the DUB inhibitor PR-619, suggesting that these two proteins are binding the OTULIN ABP via OTULIN (Fig. 1b). Thus, mass spectrometry identified SNX27 as an interactor of substrate-bound OTULIN.

To verify the interactions, we performed Western Blot (WB) after OTULIN ABP-PD from Jurkat T cell extracts. As expected, HOIP and SNX27 were co-precipitated with the OTULIN-ABP complex and association was lost when OTULIN was inhibited by PR-619 (Fig. 1c). To corroborate that SNX27 is directly binding to OTULIN, we generated OTULIN and HOIP KO Jurkat T cells using CRISPR/Cas9 technology (Supplementary Fig. 1a, b). OTULIN ABP-PD was carried out in parental and OTULIN KO Jurkat T cells and HOIP as well as SNX27 association was lost in OTULIN-deficient cells, confirming that OTULIN bridges the association of the OTULIN ABP with SNX27 (Fig. 1d). Furthermore, OTULIN and SNX27 were also co-precipitated after OTULIN ABP-PD in HOIP KO Jurkat T cells, demonstrating that OTULIN-SNX27 interaction is independent of OTULIN-HOIP association (Fig. 1e).

To confirm the SNX27–OTULIN interaction, we lentivirally reconstituted OTULIN KO Jurkat T cells with N-terminally 2xStrepII-Flag (SF)-tagged OTULIN (Supplementary Fig. 1c, d). LC-MS/MS analyses were performed after Strep-PD from mock and SF-OTULIN transduced cells. OTULIN as the bait and SNX27, but not HOIP, were identified with high significance in OTULIN-expressing versus non-expressing cells (Fig. 1f). We assume that HOIP binding to the N-terminus of OTULIN was lost during purification using the N-terminal affinity tag. Interaction of SF-OTULIN and SNX27 was verified by Western Blot after SF-PD from reconstituted Jurkat T cells (Fig. 1g).

We performed co-immunoprecipitations (IPs) of endogenous OTULIN and SNX27 in different cell lines to confirm the interaction. Anti-OTULIN IP co-precipitated SNX27 and vice versa, OTULIN was bound to SNX27 after anti-SNX27 IP in Jurkat T and HEK293T cells (Fig. 1h, Supplementary Fig. 1e). Furthermore, SNX27 was co-precipitated with OTULIN in mouse embryonal fibroblasts (MEFs) from WT but not OTULIN KO mice (Fig. 1i). Finally, OTULIN-SNX27 association was also detected in primary murine CD4 T cells after OTULIN ABP-PD or OTULIN IP (Fig. 1j). Thus, OTULIN and SNX27 interact in human and murine cell lines as well as primary cells.

**OTULIN PDZbm controls association with the SNX27 PDZ domain.** Human OTULIN encodes a putative C-terminal class I PDZ-binding motif (PDZbm: ETSL) corresponding to the consensus motif (X-S/T–X–Φ), which is highly conserved in mammals and tetrapod vertebrates (Fig. 2a, b). PDZbm-containing recycling cargos and other cytosolic proteins can be recruited to the PDZ domain of SNX27 (Fig. 2a)[17]. To validate the OTULIN-SNX27 interaction via the canonical PDZbm-PDZ interface in

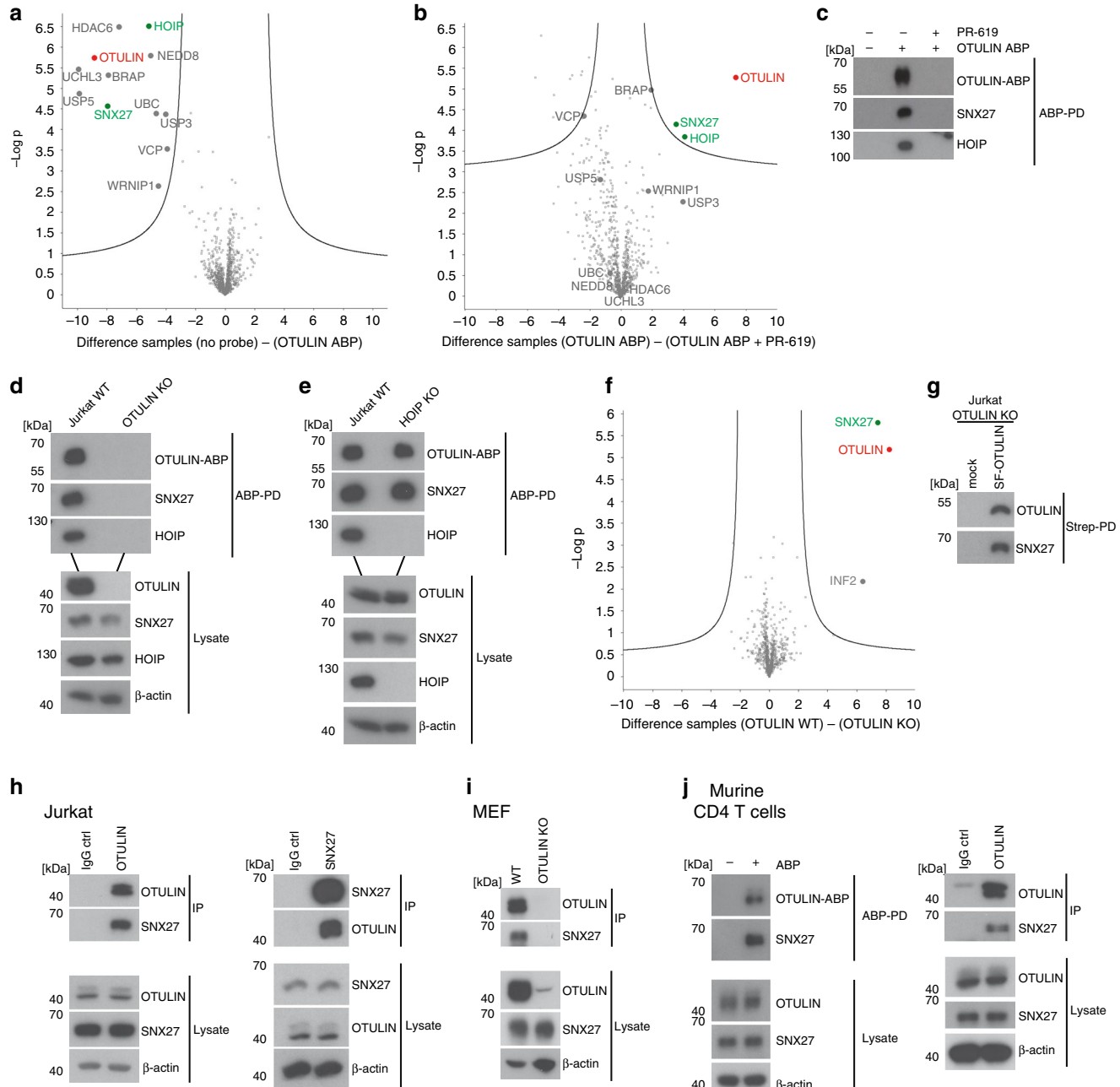

**Fig. 1** Identification of SNX27 as an OTULIN interactor. **a** Selective protein enrichment after OTULIN ABP-PD. Volcano plot illustrating the enrichment of proteins identified by LC-MS/MS after PD in the presence or absence of OTULIN ABP. Curves depict significant enrichment or depletion, respectively. **b** Selective depletion of proteins from OTULIN ABP by PR-619 treatment. Volcano plot demonstrates loss of protein binding in samples treated with PR-619 before OTULIN ABP incubation and PD. Curves depict significant enrichment or depletion, respectively. **c** OTULIN-ABP PD was performed under conditions used for LC-MS/MS and binding of HOIP and SNX27 was analyzed by WB. **d** OTULIN ABP-PD was performed from extracts of parental or OTULIN KO Jurkat T cells. Co-precipitation of HOIP and SNX27 was analyzed by WB. **e** OTULIN ABP-PD was performed from extracts of parental or HOIP KO Jurkat T cells. Co-precipitation of HOIP and SNX27 was analyzed by WB. **f** Enrichment of proteins by PD of SF-OTULIN. Volcano plot depicts significant enrichment of proteins after Strep-PD from extracts of OTULIN-deficient Jurkat T cells expressing SF-OTULIN or mock control. Curves depict significant enrichment or depletion, respectively. **g** OTULIN-SNX27 interaction in reconstituted OTULIN KO Jurkat T cells was analyzed by WB after Strep-PD. **h** OTULIN-IP and SNX27-IP from extracts of Jurkat T cells. Rabbit or mouse IgG antibodies were used as isotype controls and protein binding was analyzed by WB. **i** OTULIN-IP from extracts of WT and OTULIN KO MEFs. Co-immunoprecipitation of SNX27 was analyzed by WB. **j** ABP-PD (left) and OTULIN-IP (right) from murine primary CD4 T cell extracts. Rabbit IgG antibody was used as an isotype control. Interaction of SNX27 to murine OTULIN-ABP complex and OTULIN was assessed by WB. Source data are provided as a Source Data file

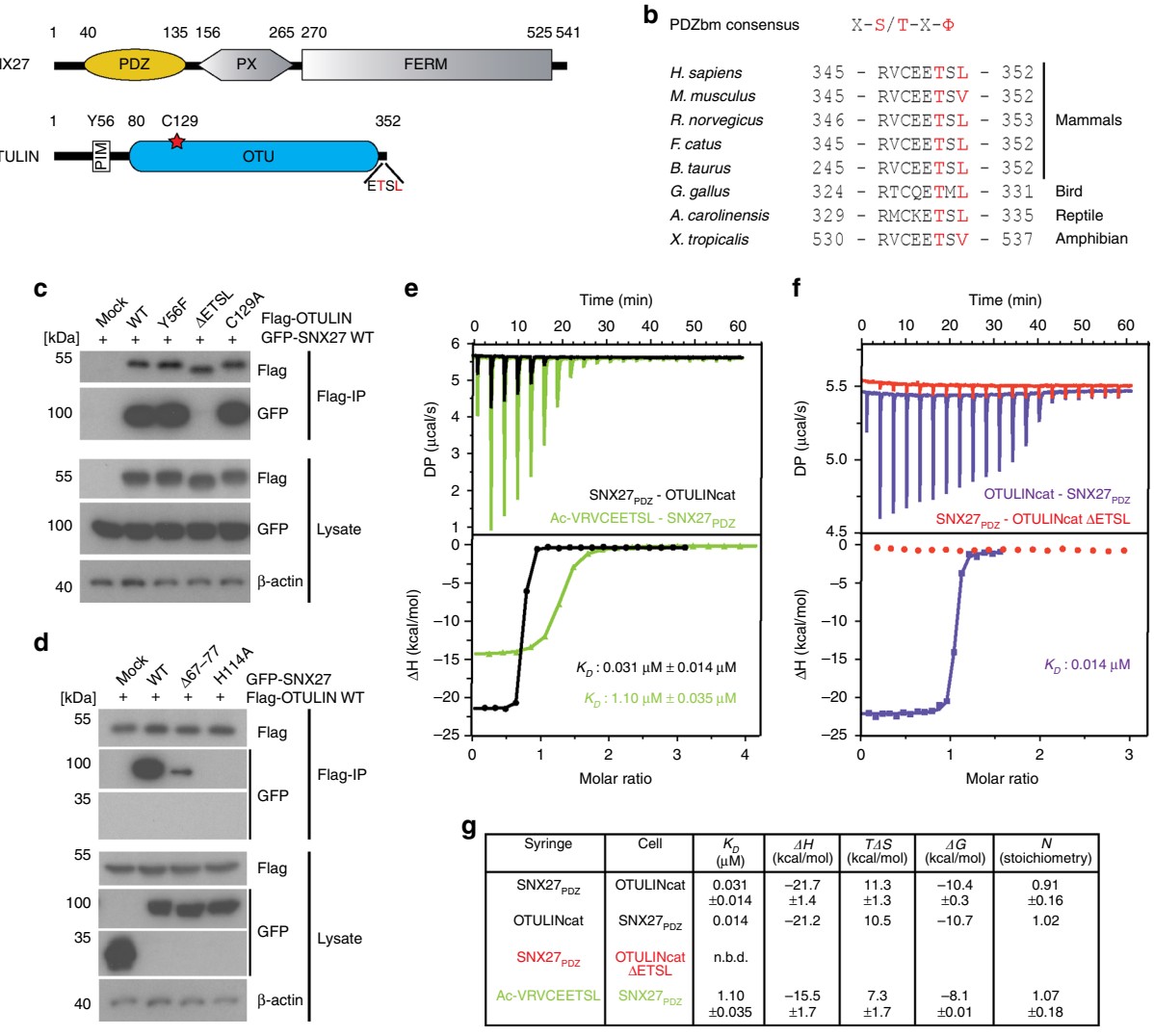

**Fig. 2** High affinity binding between OTULIN and SNX27. **a** Schematic representation of human OTULIN and SNX27 domain structure. **b** Sequence alignment and conservation of C-terminal PDZbm in OTULIN orthologues from tetrapod animals. **c**, **d** HEK293 cells were co-transfected with Flag-OTULIN and GFP-SNX27 constructs as indicated. Co-IPs were performed using anti-Flag antibody and tested for OTULIN and SNX27 interaction by WB. **e** ITC data for OTULIN PDZbm peptide (green) or OTULINcat (black) titrated against SNX27 PDZ domain. **f** ITC data for OTULINcat ΔETSL (red) and the reciprocal titration of OTULIN into SNX27 (purple). **g** Table summarizing the ITC data with mean parameters and the standard deviation derived from at least two independent experiments. Source data are provided as a Source Data file

cells, we expressed GFP-SNX27 and Flag-OTULIN constructs in HEK293 cells. Upon IP using anti-Flag antibodies, Flag-OTULIN interacted with GFP-SNX27 (Fig. 2c, d). On the side of OTULIN, deletion of the C-terminal PDZbm (ΔETSL) abrogated association with SNX27, while mutation of the catalytic site (C129A) or the PIM (Y56F) that mediates HOIP binding did not impair SNX27 binding (Fig. 2c). Conversely, mutation of His114 to Ala (H114A) in the cargo-binding interface of the SNX27 PDZ domain[21] completely abolished OTULIN binding (Fig. 2d). Interestingly, partial deletion (Δ67–77) of the β3–β4 hairpin loop (aa 67–79) that is specific for the PDZ domain of SNX27 and engages with the retromer subunit VPS26, also severely impaired the interaction of SNX27 and OTULIN (Fig. 2d)[22].

In vitro PDZbm peptides display an affinity for the PDZ domain of SNX27 with $K_D$ values typically in the μM range[17]. We performed isothermal calorimetry (ITC) and determined a $K_D$ of $1.1 \pm 0.035\,\mu M$ for the OTULIN C-terminal PDZbm peptide (aa 344–352) and the SNX27$_{PDZ}$ (aa 40–135) (Fig. 2e–g, Supplementary Table 1). With a $K_D$ of ~2 μM the PDZbm-containing peptides of DGKζ (diacylglycerol kinase ζ) and PHLPP1 (PH

Domain and Leucine Rich Repeat Protein Phosphatase 1) displayed highest affinity for SNX27$_{PDZ}$, revealing that the affinity of the isolated OTULIN PDZbm peptide was moderately higher compared to any other previously analyzed cargo peptide[17]. Next, we determined the affinity between OTULINcat (aa 80–352) and SNX27$_{PDZ}$ to get better insights into the biophysical binding properties of the OTULIN-SNX27 complex. OTULINcat and SNX27$_{PDZ}$ exhibited a $K_D$ of $0.031 \pm 0.014\,\mu M$; the affinity was more than 30-fold higher than for the OTULIN PDZbm peptide alone and to our knowledge the strongest binding ever recorded for a SNX27 PDZ domain interaction (Fig. 2e–g, Supplementary Table 1). No binding of SNX27$_{PDZ}$ to OTULINcat ΔETSL could be detected by ITC, confirming that the SNX27–OTULIN interaction relies on the C-terminal PDZbm of OTULIN (Fig. 2f, g).

To unravel the exact binding properties and to explain the high affinity of OTULIN for SNX27, we determined the structure of the catalytically inactive OTULINcat (C129A) in complex with SNX27$_{PDZ}$ at a resolution of 2.0 Å in the triclinic space group P1 (Fig. 3a, Table 1). The asymmetric unit comprised two

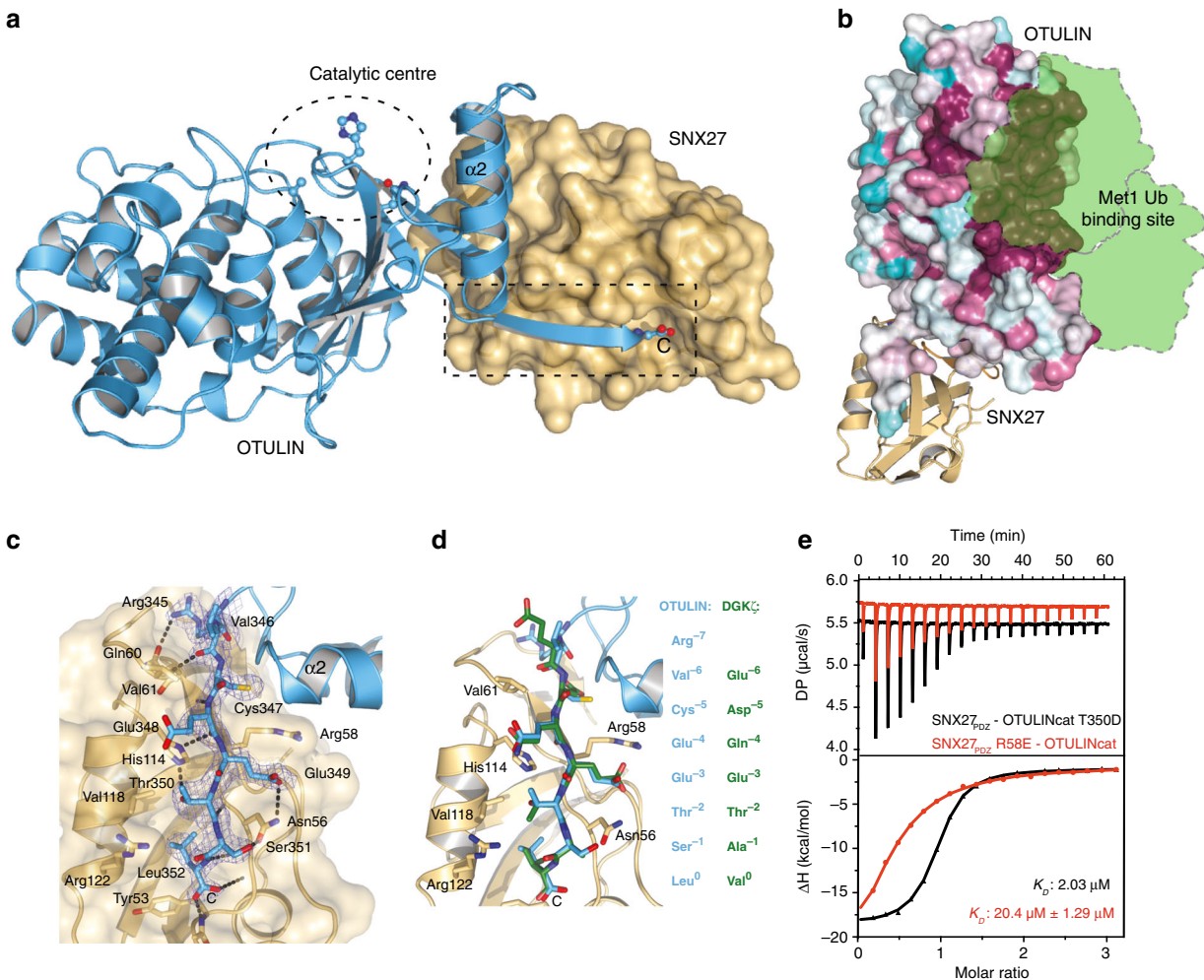

**Fig. 3** Structure of the OTULIN-SNX27 complex revealing the canonical PDZ-PDZbm interface. **a** Structure of OTULIN (blue cartoon) bound to SNX27 (brown surface). The OTULIN catalytic center and also the OTULIN PDZbm are highlighted (dotted lines). **b** Surface representation of OTULIN colored based upon conservation calculated through the Consurf webserver. The binding site for Met1 diubiquitin is shown as cartoon from superimposition of the OTULINcat Met1 diubiquitin structure (PDB: 3ZNZ). **c** Close-up view of the interaction between OTULIN PDZbm and SNX27$_{PDZ}$ rotated 90° from enclosed box in **a**. A weighted 2Fo-Fc map is shown for residues from the OTULIN PDZbm contoured at 1.1σ. Hydrogen bonds are denoted by dotted lines. **d** Additional residues from OTULIN PDZbm engage with SNX27. A canonical PDZbm cargo of SNX27 (DGKζ; PDB: 5ELQ) (green) is superimposed onto the OTULIN-SNX27 structure. Residues at each position are shown and numbered according to convention. **e** Mutation of the canonical binding site disrupts the interaction between OTULIN and SNX27. ITC data for either a single point mutation within the OTULIN PDZbm (black) or SNX27 PDZ (red) are shown

SNX27–OTULIN complexes that superimposed onto one another with an RMSD of 0.4 Å. While the C-terminus was not visible in previous structures of OTULINcat (e.g. PDB: 3ZNZ), electron density corresponding to the PDZbm could be unambiguously assigned for the entire C-terminus of OTULIN (Fig. 3a). As expected, SNX27 bound to the C-terminal tail of OTULIN comprising the PDZbm and did not affect the overall OTU fold (RMSD between apo OTULIN and OTULIN-SNX27 complex 0.9 Å). Accordingly, Met1-linked diUb could be modeled onto the OTULIN-SNX27 structure. Neither mutation of the PDZbm nor titration of excess SNX27$_{PDZ}$ altered the ability of OTULINcat to cleave Met1-linked ubiquitin chains (Fig. 3b, Supplementary Fig. 2a–c). This is in agreement with our original identification of SNX27 as a binding partner of OTULIN in complex with the Met1-diUb-based OTULIN ABP (Fig. 1a, b).

Close inspection of the C-terminal interaction interface revealed that the well-ordered OTULIN PDZbm engages with SNX27 in a canonical orientation identical to the described SNX27 peptide cargos like DGKζ (Fig. 3c, d)[17]. SNX27 His114, which was shown to be essential for cargo binding[21], forms

hydrogen bonds to the peptide backbone of the OTULIN PDZbm and also to OTULIN Thr350 (-2 position in PDZbm), explaining the strict necessity of this interaction for complex assembly (Fig. 3c). Like all class I PDZbm interactions, the C-terminal Leu352 (P0) hydrogen bonds to the backbone of Tyr53 while the aliphatic side chain occupies a hydrophobic pocket on the SNX27$_{PDZ}$ surface. Side chain hydrogen bonding occurs between OTULIN Glu349 (P-3) and Ser351 (P-1) and SNX27 Asn56 with Glu349 sandwiched between SNX27 side chains of Arg58 and Asn56. OTULIN Cys347 (P-5) forms an N-terminal helical cap that stabilizes the alpha helix (α2), which packs against the SNX27 Arg58 side chain. As mentioned, the OTULIN C-terminus is flexible and no interpretable electron density has been modeled in all the OTULIN structures solved to date. In one previous structure of OTULIN bound to Met1-diUb (PDB: 4KSJ)[7], residues 345-348 (RVCE) are observed, but form crystallographic contacts with neighboring molecules that do not resemble the conformation in complex with SNX27, suggesting that the formation of the extended ß-sheet in the OTULIN C-terminus occurs only upon binding to SNX27.

### Table 1 Data collection and refinement statistics

| | OTULINcat–SNX27$_{PDZ}$ |
|---|---|
| *Data collection* | |
| Beamline | Diamond I03 |
| Space group | P 1 |
| *a, b, c* (Å) | 54.65, 62.92, 72.02 |
| *α, β, γ* (°) | 65.13, 85.98, 85.38 |
| Wavelength (Å) | 0.96861 |
| Resolution (Å) | 42.65–2.00 (2.05–2.00) |
| $R_{merge}$ | 8.8 (53.1) |
| <I / σI > | 5.7 (1.8) |
| CC (1/2) | 0.99 (0.38) |
| Completeness (%) | 95.6 (94.2) |
| Redundancy | 1.7 (1.8) |
| *Refinement* | |
| Resolution (Å) | 42.66–2.00 |
| No. reflections | 55,971 |
| $R_{work}/R_{free}$ | 21.5/25.3 |
| No. atoms | |
| Protein | 5674 |
| Ligand / ion | 84 |
| Water | 268 |
| *B-factors* | |
| Wilson B | 34.01 |
| Protein | 52.63 |
| Ligand/ion | 88.02 |
| Water | 59.9 |
| R.m.s. deviations | |
| Bond length (Å) | 0.003 |
| Angels (°) | 0.581 |
| Ramachandran statistics (outliers, allowed, favored) | 0.0, 2.5, 97.5 |

Numbers in parentheses are for the highest resolution bin

Interestingly, the canonical PDZbm of OTULIN extends further than the typically reported P-5 position[17]. OTULIN Arg345 at P-7 hydrogen bonds to the side chain oxygen of SNX27 Gln60 and backbone hydrogen bonding occurs between Val346 (P-6) and SNX27 Val61. To validate the importance of the canonical PDZ-PDZbm binding site we mutated residues in SNX27 and OTULIN that mediate this interface. Mutation of either OTULIN Thr350 to Asp (T350D) or of SNX27 Arg58 to Glu (R58E) significantly reduced the association of SNX27$_{PDZ}$ with OTULINcat to 2 μM or 20 μM, respectively (Fig. 3e).

**Second interface mediates high-affinity OTULIN-SNX27 binding**. The structure revealed a second OTULIN-SNX27 interface besides the canonical PDZ-PDZbm interaction (Fig. 4a, b). Within this interface residues of the OTU domain are in proximity to the unique extension that forms the exposed β3–β4 hairpin loop (aa 67–79) in the SNX27 PDZ domain, which was shown to also engage with the retromer subunit VPS26A[22]. As shown above, deletion of the β3–β4 extension strongly diminished binding of OTULIN to SNX27 (Fig. 2d). The interface is composed of polar and electrostatic interactions with the side chain carboxyl groups of OTULIN Asp90, Asp87, and Glu85 hydrogen bonding to SNX27 Arg100 and Arg68. Of note, the negatively charged amino acids in the secondary SNX27 binding surface of OTULIN are highly conserved (Fig. 4c). Further, OTULIN Arg345 close to the PDZbm forms stabilizing interactions between OTULIN residues Glu85 and Glu209 in addition to hydrogen bonding to SNX27 Gln60. Finally, OTULIN Glu209 hydrogen bonds to the amide backbone of SNX27 Gly64.

To validate whether this second interface outside the classical PDZbm binding region is contributing to the OTULIN-SNX27 interaction, we generated structure-guided mutations and determined affinities by ITC. SNX27$_{PDZ}$ mutations of Gly64 to Glu (G64E) or Gly65 to Ala (G65A) modestly reduced the binding affinity to OTULIN by ~11 fold (0.34 μM) and ~3 fold (0.093 μM), respectively (Fig. 4d, Supplementary Fig. 3a). Similar effects were measured with the charged reversion of Arg100 to Glu (R100E) (0.26 μM, ~8 fold). Combination of both mutations that would abrogate the second binding site (G64E, R100E) reduced the affinity to OTULIN to 1.1 μM (~35 fold), which is equivalent to the dissociation constant measured with just OTULIN PDZbm peptide and SNX27$_{PDZ}$ (1.1 μM) (Fig. 2e). In line, the double mutation G64E, R100E severely reduced but did not abolish the binding of full length GFP-SNX27 to HA-OTULIN in HEK293 cells (Fig. 4e). Reciprocal mutations on the side of OTULIN had similar outcomes. Modest to severe effects on binding to SNX27$_{PDZ}$ were obtained by single charge reversion mutations of OTULIN Glu209 to Arg (E209R; 0.05 μM, ~1.6 fold), Asp90 to Arg (D90R; 0.12 μM, ~4 fold), Asp87 to Arg (D87R; 0.31 μM, ~10-fold), and Glu85 to Arg (E85R; 0.84 μM, ~27 fold) (Fig. 4d, Supplementary Fig. 3b). Again, the combination of E85R/D87R had the strongest effect and reduced the affinity to 2.07 μM (~68 fold). Furthermore, double mutation E85R/D87R in full length HA-OTULIN strongly impaired the binding to GFP-SNX27 in HEK293 cells (Fig. 4f). No effects on OTULIN catalytic activity was observed by the E85R/D87R mutation in the secondary SNX27-binding site (Supplementary Fig. 2b, c). Thus, while the canonical PDZ-PDZbm binding interface is essential for the association, the second interface composed of the SNX27 PDZ β-hairpin loop and residues Glu85 and Asp87 in OTULIN contributes to the high affinity of SNX27 and OTULIN.

Using two mass spectrometry approaches, we defined SNX27 as an OTULIN interactor, but despite the C-terminal class I PDZbm consensus in OTULIN, no other PDZ domain-containing proteins were identified (Fig. 1a, f). Since the sequence around the β3–β4 hairpin loop insertion is found exclusively in the PDZ domain of SNX27 (Supplementary Fig. 3c)[22], we wanted to investigate if this second interface might explain the high selectivity of OTULIN for SNX27. Superimposition of structurally related PDZ domains onto SNX27 revealed potential clashes between OTULIN and the PDZ domains of NHERF2, SHANK1, and LNX2 (Fig. 4g). Other PDZ domains including Syntenin1 PDZ-1 and RhoGEF could accommodate the extended interface from OTULIN, but they apparently lack additional stabilizing residues from the second interface. We purified PDZ domains of several proteins and determined their affinity to OTULINcat by ITC (Supplementary Fig. 3d, Supplementary Table 1). While PDZ domains of MUPP1 and PSD95 (PDZ 3) did not show any detectable binding to OTULINcat, measured $K_D$ values of other PDZs ranged between ~22 and 26 μM for PDZs from SHANK1, SHANK2, and SHANK3 to 1710 μM for Syntenin1 PDZ1, demonstrating at least 700-fold lower affinity for any other PDZ domain tested in the OTULINcat binding assay. Thus, other PDZ domains either clash with the OTULIN OTU domain or lack the second interaction surface leading to absence of binding or much lower affinities, explaining why in cells only SNX27 was detected as an OTULIN interactor.

**OTULIN counteracts binding of SNX27 to the VPS26A ret-romer**. Despite the robust identification of SNX27 as an OTULIN interactor in cells, mass spectrometry did not reveal any evidence that OTULIN associated directly or indirectly with other retromer complex subunits such as VPS26, VPS35, or VPS29[16]. VPS26A directly binds to the β3–β4 hairpin loop, unique to the SNX27 PDZ domain (PDB: 4P2A)[22]. In the crystal structure of SNX27 PDZ bound to VPS26A the canonical cargo binding site in the

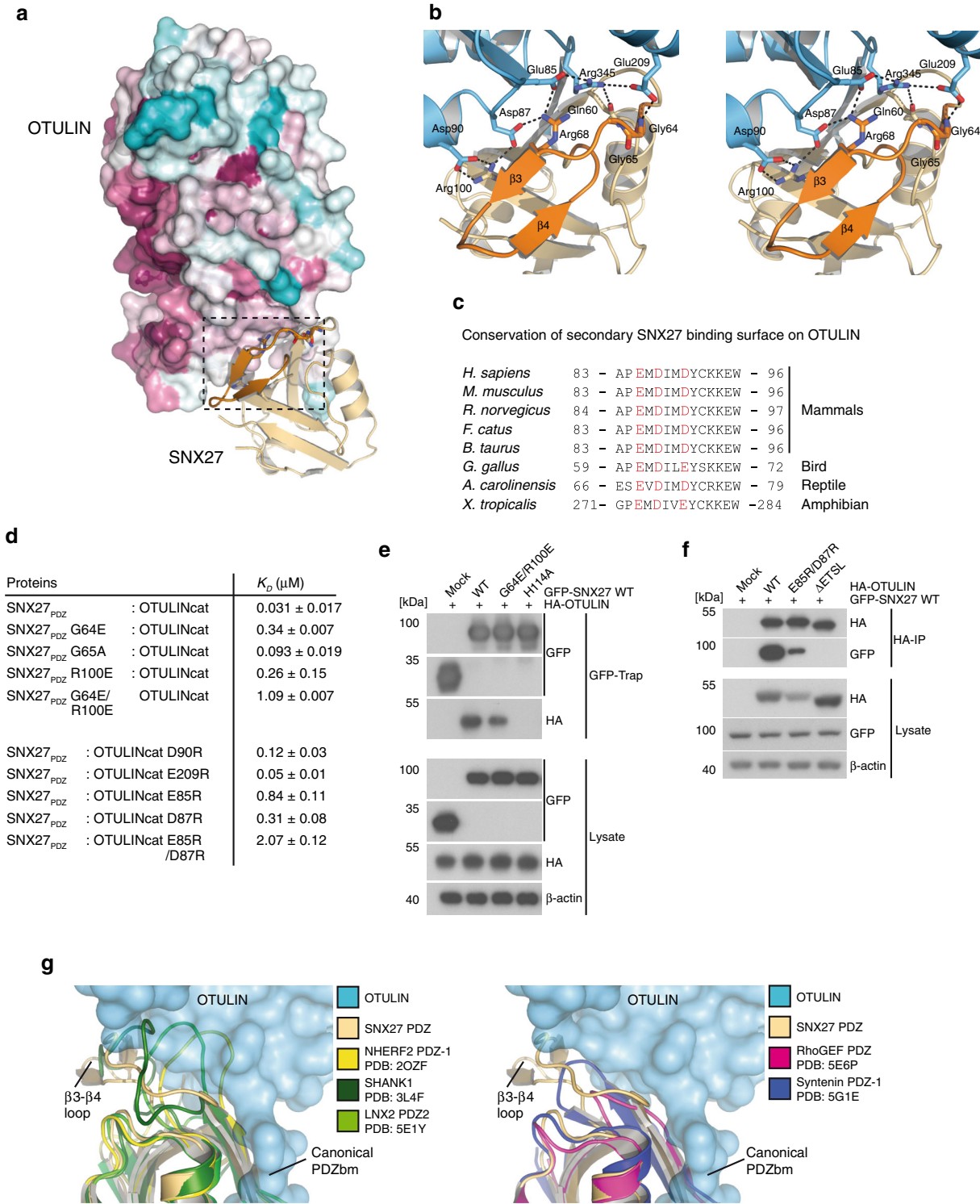

**Fig. 4** Secondary interface mediates high affinity interaction and selectivity between OTULIN and SNX27. **a** Surface representation of OTULIN colored based upon conservation calculated through the Consurf webserver. Box encloses the second SNX27 interface. **b** Close-up stereo view of the second interface of OTULIN from **a**. The unique β3–β4 hairpin insertion within the SNX27 PDZ domain is shown in orange. Hydrogen bonds between residues are shown with dotted lines. **c** Sequence alignment and conservation of second SNX27 binding surface in OTULIN orthologues from tetrapod animals. Conserved polar and electrostatic amino acids involved in binding to SNX27 are depicted in red. **d** Summary of SNX27–OTULIN affinities ($K_D$) determined by ITC for mutations in the secondary interface. **e** HEK293 cells were co-transfected with HA-OTULIN and GFP-SNX27 WT or mutants that carry missense mutations in the canonical cargo binding site (H114A) or in the second OTULIN interface (G64E/R100E). Binding was analyzed after GFP-Traps by WB. **f** HEK293 cells were co-transfected with GFP-SNX27 and HA-OTULIN constructs. OTULIN mutants either lack the C-terminal PDZbm (ΔETSL) or carry missense mutations in the second SNX27 interface (E85R/D87R). SNX27 binding was analyzed after HA-IP by WB. **g** Superimposition of structurally related PDZ domains onto SNX27–OTULIN structure. All closely related PDZ domains lack the SNX27 β3–β4 hairpin insertion. PDZ domains of SHANK1, NHERF1, and LNX2 contain a larger insertion that may clash with the second interface on OTULIN (left). PDZs of RhoGEF and Syntenin1 contain a short loop at the equivalent position that would not clash with the second interface of OTULIN (right). Source data are provided as a Source Data file

PDZ domain is accessible for the C-terminal OTULIN PDZbm (Fig. 5a). However, VPS26A and the second interface of OTULIN are in close proximity to the SNX27 β3–β4 hairpin (Figs. 5b and 4b). OTULIN contacts Gly64 and Gly65 preceding the hairpin, but also accommodates a salt bridge to Arg68 within the β3-sheet of the SNX27 PDZ domain (Fig. 4b). Gly65 and Arg68 in SNX27 are also involved in binding to VPS26A[22] (Fig. 5b). Moreover, simultaneous binding of OTULIN and VPS26A to SNX27 is incompatible with the structural constraints of these complexes as OTULIN would clash with VPS26A when both are bound to SNX27 PDZ (Fig. 5a, c). Thus, besides the canonical cargo binding surface on SNX27, OTULIN and VPS26A utilize partially overlapping binding surfaces in the β3-β4 hairpin loop of SNX27 (Fig. 5d). Using ITC we measured a $K_D$ of 27.5 μM for binding of VPS26A and SNX27 PDZ, which is an almost 1000-fold lower affinity compared with OTULIN and SNX27 (Fig. 5e, Supplementary Table 1). In line, size exclusion chromatography using purified proteins revealed a complex with 1:1 stoichiometry between OTULINcat and SNX27 PDZ (Supplementary Fig. 4). While no OTULIN-VPS26A complex was formed, SNX27 PDZ only slightly trailed towards VPS26A-containing fractions, validating the rather loose association of the two purified proteins, which most likely results from high $K_{on}/K_{off}$ rates. Thus, the molecular architectures and association data of the SNX27-VPS26A and SNX27–OTULIN complexes provide clear evidence that binding of OTULIN and VPS26A to SNX27 is mutually exclusive. Based on the structural information OTULIN cannot be a cargo of SNX27, but in light of its high affinity should rather antagonize SNX27 cargo loading and retromer assembly.

To explore the relevance of the in vitro findings in cells we co-expressed GFP-SNX27 together with Flag-VPS26A in the presence or absence of HA-OTULIN (Fig. 5f). Indeed, GFP-Trap co-precipitated VPS26A in complex with SNX27, but upon expression of OTULIN, SNX27 binding to VPS26A was lost and only SNX27–OTULIN complexes were detectable. Similarly, GFP-SNX27 precipitated endogenous VPS26A and VPS35 (Fig. 5g). SNX27 binding to both retromer subunits was strongly reduced upon expression of HA-OTULIN and this antagonistic effect strictly relied on the presence of the C-terminal PDZbm. Also, OTULIN active site mutation C129A did not affect its ability to remove the VPS26/VPS35 retromer from SNX27.

**OTULIN antagonizes endosomal localization of SNX27.** To follow the association of cellular OTULIN and SNX27 and their potential distribution in LUBAC and/or retromer complexes, we performed size exclusion chromatography using extracts of Jurkat T cells (Fig. 6a, Supplementary Fig. 5a). As observed previously, HOIP-containing LUBAC complex peaks in fractions 4–7 with an apparent MW of >440 kDa[9,23,24]. OTULIN largely exists in a HOIP-unbound fraction and LUBAC association was enhanced by OTULIN dephosphorylation[9]. SNX27 did not significantly co-migrate with VPS26 and VPS35 retromer subunits, which predominantly eluted in fractions 8–11 correlating with a MW of ~150–440 kDa (Fig. 6a). Importantly, OTULIN eluted together with SNX27 in fractions 12–14 at an apparent MW between 44 kDa and 158 kDa. Thus, peak elution profiles of OTULIN (~42 kDa) and SNX27 (~61 kDa) correspond to a complex with a molecular mass of ~100 kDa, which would be the expected size for an OTULIN-SNX27 heterodimer as seen in the crystal structure (Fig. 3a). To verify that OTULIN and SNX27 are associating in these fractions, we performed the same size exclusion chromatography using extracts of OTULIN or SNX27 KO Jurkat T cells (Supplementary Figs. 1a and 6a). Indeed, elution profiles of OTULIN and SNX27 were shifted to the lower MW by 1 and 2 fractions in SNX27 and OTULIN KO cells,

respectively (Fig. 6a). Whereas OTULIN peaked in fractions 14-15, SNX27 maximum was between fractions 13–15, which roughly corresponds to the expected elution of the monomers. Elution profiles of HOIP or VPS26/VPS35 were not altered in OTULIN or SNX27 KO Jurkat T cells (Supplementary Fig. 5b, c). Also, despite the antagonism of OTULIN and VPS26A for SNX27 binding, SNX27 was not shifted to VPS26/VPS35 retromer-containing fractions in OTULIN KO cells (Fig. 6a, Supplementary Fig. 5b). Thus, gel filtration provides evidence that endogenous OTULIN and SNX27 form a stable cytosolic complex that is not associated with LUBAC or the VPS26-containing recycling retromer complexes.

To visualize the interaction and cellular localization of SNX27–OTULIN complexes, we expressed GFP-SNX27 and RFP-OTULIN in HEK293 cells and performed confocal fluorescence microscopy. GFP-SNX27 was present in the cytosol but highest fluorescence intensities were found in subcellular substructures positive for EEA1 staining, which is consistent with previous reports showing accumulation of SNX27 on EEA1-positive early endosomes (Fig. 6b)[25,26]. Lentiviral co-expression of RFP-OTULIN WT or inactive C129A led to an evenly distributed co-localization of SNX27 with OTULIN throughout the cytoplasm and redistribution of SNX27 away from the endosomal compartment (Fig. 6c). Removal of the PDZbm in OTULIN ΔETSL prevented redistribution of SNX27 to the cytoplasm and OTULIN was slightly more localizing to the nucleus. The OTULIN-driven impairment of endogenous SNX27 to co-localize with EEA1-positive endosomes was corroborated in HEK293 cells subjected to lentiviral expression of OTULIN WT and C129A but not OTULIN ΔETSL (Fig. 6d). Finally, we found a slight but significant increase in the endosomal localization of endogenous SNX27 to EEA1-positive endosomes in OTULIN KO HEK293 cells compared to WT controls (Fig. 6e). These data further confirm a model whereby OTULIN extracts SNX27 from endosomal-associated retromer recycling complexes.

**SNX27 deficiency does not affect OTULIN and LUBAC functions.** OTULIN associates with HOIP through the N-terminal PIM and with SNX27 via the C-terminal PDZbm. To address a potential role of SNX27 in the regulation of LUBAC and Met-1 Ub chain homeostasis by OTULIN, we asked if HOIP, OTULIN and SNX27 could assemble into a ternary complex. We co-expressed GFP-SNX27, Flag-OTULIN and HA-HOIP in HEK293 cells and pulled-down SNX27 by GFP-Traps (Fig. 7a). Whereas OTULIN was bound to SNX27 in the absence of HOIP, HOIP co-precipitated with SNX27 only in the presence of OTULIN. Moreover, HOIP binding to SNX27 strictly relied on both the SNX27 recruitment surface (PDZbm mutant ΔETSL) and the functional HOIP binding motif (PIM mutant Y56F) in OTULIN. Thus, the binding of SNX27 and HOIP to non-overlapping interfaces on OTULIN allows all three proteins to form a ternary complex in cells.

Next, we investigated if SNX27 deficiency affects the function of OTULIN in the homeostatic control of Met1-linked ubiquitin chains and as an antagonist of LUBAC. For this, a panel of CRISPR/Cas9-generated HEK293 and Jurkat T cell clones that lacked either OTULIN or SNX27 expression were analyzed for HOIP expression and homeostatic synthesis of Met1-linked ubiquitin chains (Fig. 7b, Supplementary Fig. 6b). As previously described, the LUBAC subunit HOIP was downregulated in the absence of OTULIN[13], but in SNX27-deficient cells HOIP expression was not decreased. HOIP amounts were even increased in SNX27-deficient HEK293 cells, but this effect was not seen in SNX27 KO Jurkat T cells. SNX27 expression was slightly decreased in OTULIN KO Jurkat T cells, but again this

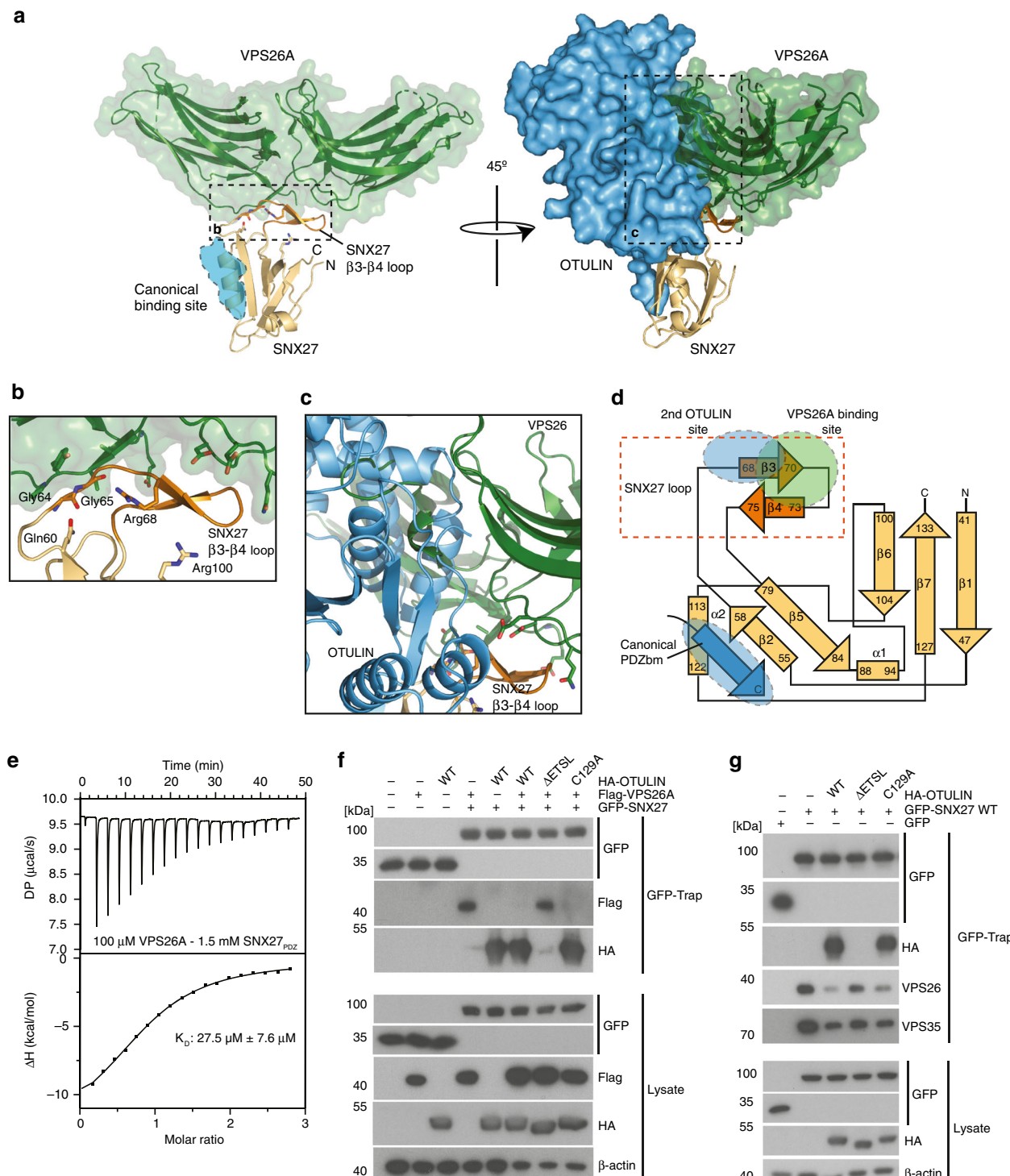

**Fig. 5** OTULIN competes with VPS26 for SNX27 binding. **a** SNX27 utilizes residues from the β3–β4 insertion to engage with the retromer complex. Structure of SNX27 bound to VPS26A (green surface) (PDB: 4P2A). The canonical PDZbm of OTULIN is shown as a blue cartoon. Rotation of the complex by 45° with superimposition of the OTULIN-SNX27 and VPS26A-SNX27 structures onto SNX27. **b** Close-up view of **a**, with residues from the SNX27 β3–β4 hairpin that bind to VPS26A shown. **c** Close-up view of **a** revealing extensive clashes between OTULIN and VPS26A through overlapping binding sites on the β3–β4 hairpin. **d** Topology diagram of SNX27 showing the canonical PDZbm binding site and the VPS26A and OTULIN binding sites on the β3–β4 hairpin. **e** ITC data for SNX27 PDZ domain titrated against VPS26A. **f** HEK293 cells were co-transfected with GFP-SNX27, Flag-VPS26A, and HA-OTULIN constructs as indicated. GFP-SNX27 was precipitated using GFP-Traps and tested for simultaneous interactions with VPS26A and OTULIN by WB. **g** HEK293 cells were co-transfected with GFP-SNX27 and HA-OTULIN constructs as indicated. Following GFP-Trap precipitation, binding of GFP-SNX27 to HA-OTULIN and effects on interaction of GFP-SNX27 with the endogenous retromer subunits VPS26 and VPS35 were monitored by WB. Source data are provided as a Source Data file

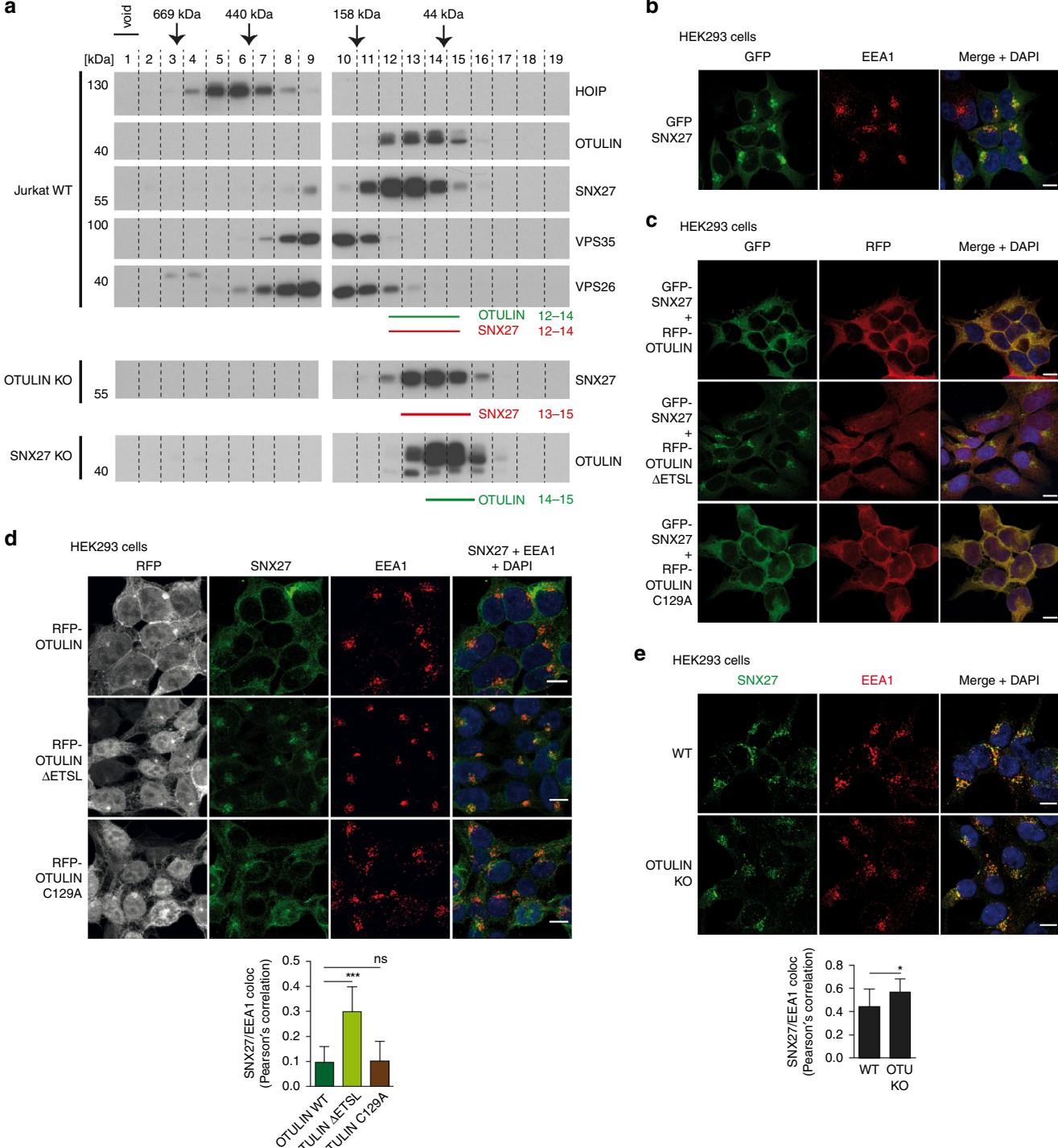

**Fig. 6** Binding of OTULIN antagonizes SNX27 association with early endosomes. **a** Co-elution of endogenous OTULIN and SNX27 was analyzed by size exclusion chromatography. Upper panels: cell lysates of parental Jurkat T cells were fractionated using a Superdex 200 column and elution profiles of endogenous proteins (OTULIN, SNX27, HOIP, VPS35, and VPS26) were determined by WB. Lower panels: determination of elution profiles of endogenous SNX27 and OTULIN in OTULIN and SNX27 KO Jurkat T cells, respectively. Peak elution of molecular weight standards is depicted at the top. **b** HEK293 cells virally transduced with GFP-SNX27 (green) were stained for early endosomes using anti-EEA1 antibody (red) and co-localization was analyzed by confocal microscopy. **c** HEK293 cells were co-transduced with GFP-SNX27 (green) and RFP-OTULIN WT, ΔETSL or C129A (red). Localization of proteins was analyzed by confocal fluorescence microscopy. **d** HEK293 cells virally transduced with RFP-OTULIN WT, ΔETSL or C129A (gray) were stained for endogenous SNX27 (green) and EEA1 (red) and localization was analyzed by confocal fluorescence microscopy. **e** WT or OTULIN KO HEK293 cells were stained for endogenous SNX27 (green) and EEA1 (red) and localization was analyzed by confocal fluorescence microscopy. Co-localization in **d** and **e** was quantified by determining Pearson's correlation using at least 12 pictures and imaging more than 100 cells for each condition. Graphs depict the mean ± SD. Two-tailed p-values: ns not significant, *$p \leq 0.05$, ***$p \leq 0.001$ by unpaired t-test. Scale bars: 10 μm. Source data are provided as a Source Data file

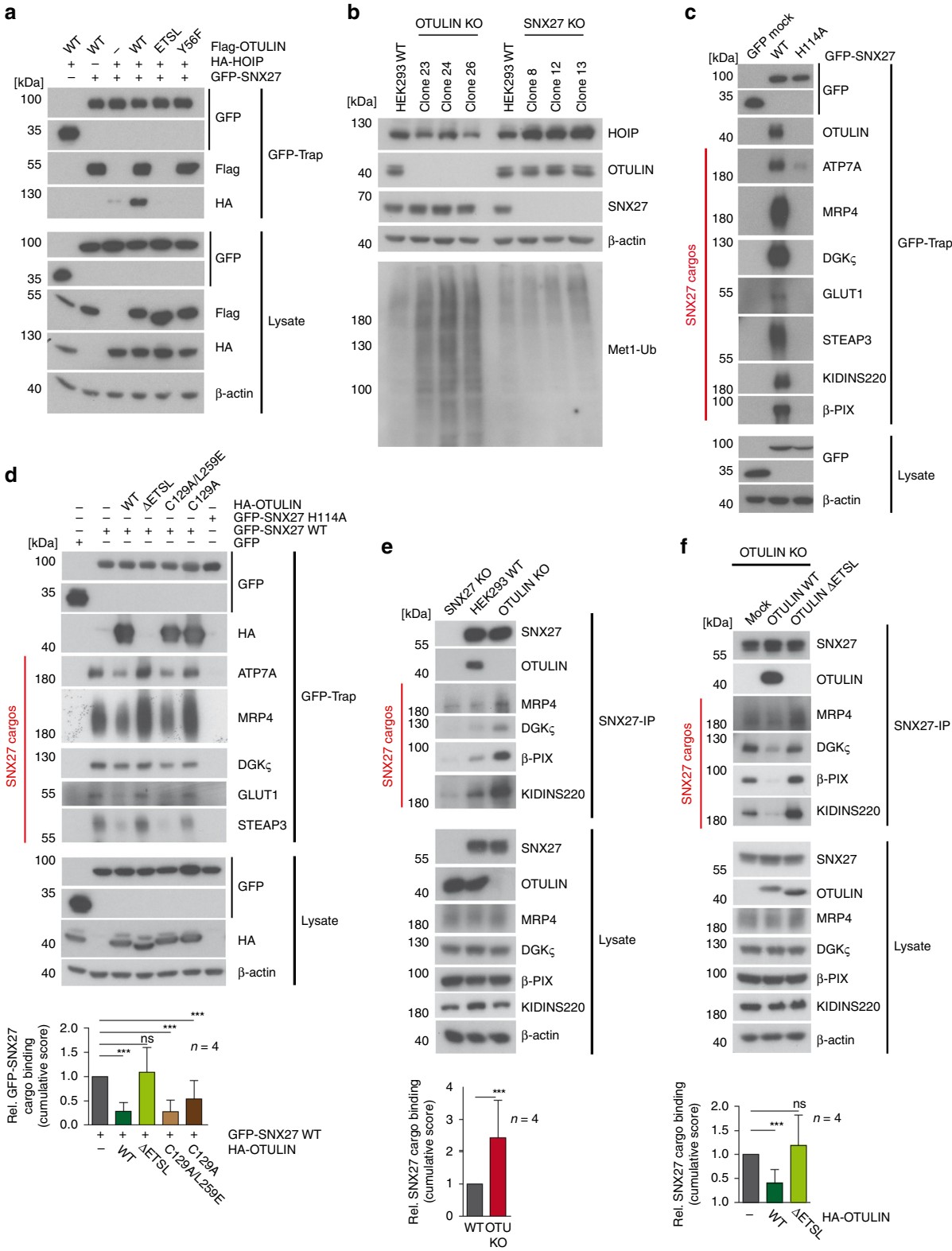

was not observed in HEK293 cells. In line with the homeostatic control of ubiquitin chain synthesis by OTULIN, Met1-linked polyUb accumulated in OTULIN KO clones from both cell lines, but Met1-linked ubiquitin chain synthesis was not altered in SNX27 KO clones (Fig. 7b, Supplementary Fig. 6b).

Since OTULIN can act as a negative regulator of NF-κB signaling[5,13], we wanted to determine if absence of SNX27 would

affect NF-κB signaling in Jurkat T cells in response to the inflammatory cytokine TNFα or antigenic TCR/CD28 co-stimulation. As expected, while HOIP deficiency impaired phosphorylation and degradation of IκBα in response to TNFα, absence of OTULIN augmented NF-κB signaling (Supplementary Fig. 6c). Neither reduction nor genetic ablation of SNX27 prompted alterations in TNFα- or CD3/CD28-induced NF-κB

**Fig. 7** OTULIN counteracts SNX27 cargo loading. **a** Formation of a ternary HOIP-OTULIN-SNX27 complex was analyzed after co-expression of GFP-SNX27, HA-HOIP, and Flag-OTULIN in HEK293 cells. Ternary complex via Flag-OTULIN was verified using OTULIN Y56F (PIM mutation) and ΔETSL (PDZbm deletion) mutants. GFP-SNX27 was precipitated by GFP-Traps and association of OTULIN and HOIP was analyzed by WB. **b** SNX27 deficiency does not affect accumulation of Met1-ubiquitin chains. Extracts of WT, OTULIN-deficient, or SNX27-deficient HEK293 cells were analyzed for the abundance of Met1-ubiquitin chains and the expression of HOIP, OTULIN, and SNX27 by WB. **c** GFP or GFP-SNX27 constructs (WT or H114A) were expressed in HEK293 cells and cargo loading was analyzed by GFP-Traps by WB. **d** GFP-SNX27 WT was expressed in HEK293 cells alone or in the presence of HA-OTULIN WT, ΔETSL (PDZbm deletion), C129A (catalytically inactive), or C129A/L259E (catalytically inactive ubiquitin-binding mutant) and cargo loading was analyzed by GFP-Traps from cell lysates by WB. **e** Cargo-loading to endogenous SNX27 was assessed after anti-SNX27-IP in parental HEK293 cells as well as SNX27 KO or OTULIN KO cells. Binding of SNX27 to cargos and OTULIN was analyzed by WB. **f** Cargo-loading onto endogenous SNX27 was assessed by WB after anti-SNX27-IP in OTULIN KO HEK293 cells reconstituted with mock, OTULIN WT or OTULIN ΔETSL. Binding of each cargo to GFP-SNX27 (**d**) or SNX27 (**e, f**) was quantified from four independent experiments (see Supplementary Fig. 7a, b, e). A cumulative score for relative binding of all cargos to SNX27 was calculated and depicted as mean ± SD below the WB. Two-tailed $p$-values: ns not significant, ***$p \leq 0.001$ by unpaired $t$-test. Source data are provided as a Source Data file

---

signaling (Supplementary Fig. 6d, e). Previous results indicated that SNX27 may negatively affect ERK phosphorylation in response to TCR/CD28 stimulation in Jurkat T cells[27], but we did not observe significant changes in ERK phosphorylation in SNX27 KO Jurkat T cells (Supplementary Fig. 6e). Furthermore, OTULIN was shown to bind DVL2 to modulate WNT signaling and SNX27 can control surface expression of frizzled receptors[7,28]. To address a potential impact of SNX27 on WNT signaling, we tested for β-catenin stabilization following WNT3a stimulation in SNX27 KO Jurkat T cells (Supplementary Fig. 6f). Again, no significant differences in the extent of β-catenin stabilization were detected in the absence of SNX27. Taken together, lack of SNX27 does not alter homeostatic control of Met1-linked ubiquitin chains by OTULIN and does not impact on NF-κB and WNT signaling pathways, suggesting that OTULIN-SNX27 binding is not controlling signaling by impacting on ubiquitin homeostasis.

**OTULIN impairs SNX27 cargo binding and trafficking.** To investigate if OTULIN regulates cargo loading onto SNX27 and may thereby be involved in endosomal recycling, we first determined GFP-SNX27 binding to a panel of previously identified SNX27 cargos in HEK293 cells[18]. As expected, the SNX27 PDZ binders ATP7A, MRP4, DGKζ, GLUT1, STEAP3 KIDINS220, and β-PIX co-precipitated with GFP-SNX27 (Fig. 7c). Cargo binding was strongly reduced by the mutation H114A in the canonical PDZbm binding surface. Importantly, when we co-expressed HA-OTULIN, overall association of cargos to SNX27 were severely diminished (Fig. 7d, Supplementary Fig. 7a). OTULIN ΔETSL mutant was unable to compete for cargo binding, underscoring that the effect is relying on the presence of the C-terminal class I PDZbm and thus SNX27 association. Interestingly, catalytically inactive OTULIN C129A still antagonized cargo binding, but with slightly less efficiency compared to OTULIN WT. However, in cells expressing the catalytically inactive OTULIN C129A mutant, Met1-linked polyUb chains accumulate and are stably bound by OTULIN C129A ($K_D$ of 150 nM), which may obstruct its ability to antagonize SNX27 cargo binding[5]. Indeed, the double mutant OTULIN C129A/L259E, which also fails to bind Met1-linked ubiquitin chains, associated with SNX27 and antagonized the ability of SNX27 to recruit cargos to the same extend as OTULIN WT (Fig. 7d).

To corroborate the antagonistic effect of OTULIN, we performed IPs and compared recruitment of cargos in parental SNX27 KO and OTULIN KO HEK293 cells to endogenous SNX27 (Fig. 7e, Supplementary Fig. 7b). Just like the OTULIN-SNX27 association, binding of the cargos MRP4, DGKζ, β-PIX, and KIDINS220 was not seen in SNX27 KO cells, confirming the specificity of the endogenous co-IP. Importantly, loss of

SNX27–OTULIN binding augmented association of SNX27 to the cargos in OTULIN-deficient cells, while cargo expression was not altered in the KO cells. To validate that loss of OTULIN association caused the enhancement of SNX27-cargo recruitment, we rescued OTULIN expression by viral transduction, which led to homogenous and equivalent expression of OTULIN WT and ΔETSL in OTULIN KO HEK293 cells (Supplementary Fig. 7c, d). Whereas OTULIN WT was recruited to SNX27 and impaired association of MRP4, DGKζ, β-PIX, and KIDINS220, OTULIN ΔETSL failed to compete for cargo binding, confirming that of the OTULIN-SNX27 interaction is responsible for counteracting cargo recruitment to SNX27 (Fig. 7f, Supplementary Fig. 7e).

Next, we investigated whether modulation of cellular OTULIN-SNX27 interaction affects endosome-to-plasma membrane trafficking in U2OS and HeLa cells, which have been widely used to monitor recycling of SNX27 cargos such as endogenous glucose transporter 1 (GLUT1) or the trafficking reporter GFP-SLC1A4[18,22,29]. We determined GLUT1 levels on the cell surface of U2OS and HeLa cells by flow cytometry after transfection of mock, OTULIN or OTULIN ΔETSL together with the surface marker ΔCD2 (Fig. 8a, Supplementary Fig. 8a). No alterations in GLUT1 surface expression were detected in untransfected cells, which lack the co-expressed marker ΔCD2 (ΔCD2-low cells, middle panels). However, transfected ΔCD2-high U2OS and HeLa cells overexpressing OTULIN showed a marked reduction in GLUT1 amounts on the cell surface (right panels). Again, removal of the OTULIN PDZbm (ΔETSL) abolished the negative effect of OTULIN on GLUT1 levels on the plasma membrane. Next, we analyzed sorting of virally transduced GFP-SLC1A4, an amino acid transporter on the cell surface that requires the SNX27-retromer for recycling and that accumulates in lysosomes if SNX27 is depleted[29]. In virally transduced U2OS and HeLa cells expressing GFP-SLC1A4, co-expression of OTULIN WT but not OTULIN ΔETSL led to a redistribution of GFP-SLC1A4 from the cell surface and to an accumulation in LAMP1-positive lysosomes (Fig. 8b, c, Supplementary Fig. 8b, c). Overall, these data demonstrate that competition for SNX27 binding caused lysosomal mis-sorting of SLC1A4.

Since clonal selection and adaptation may perturb trafficking in KO cells, we used siRNA knock-down to analyze cell surface expression of endogenous GLUT1 in U2OS cells by FACS. While knock-down of SNX27 or PDZbm-containing ANKRD50, an essential component of the SNX27-retromer recycling complex[29], decreased GLUT1 amounts on the cell surface, OTULIN depletion exerted the opposite effect by increasing GLUT1 at the plasma membrane (Fig. 8d–f, Supplementary Fig. 8d–f). Knock-down of the PDZbm-containing SNX27 cargo DGKζ did not significantly affect GLUT1 surface levels. Importantly, the combined SNX27/OTULIN knock-down restored GLUT1 surface expression compared to SNX27 single knock-down cells (Fig. 8d–f). To

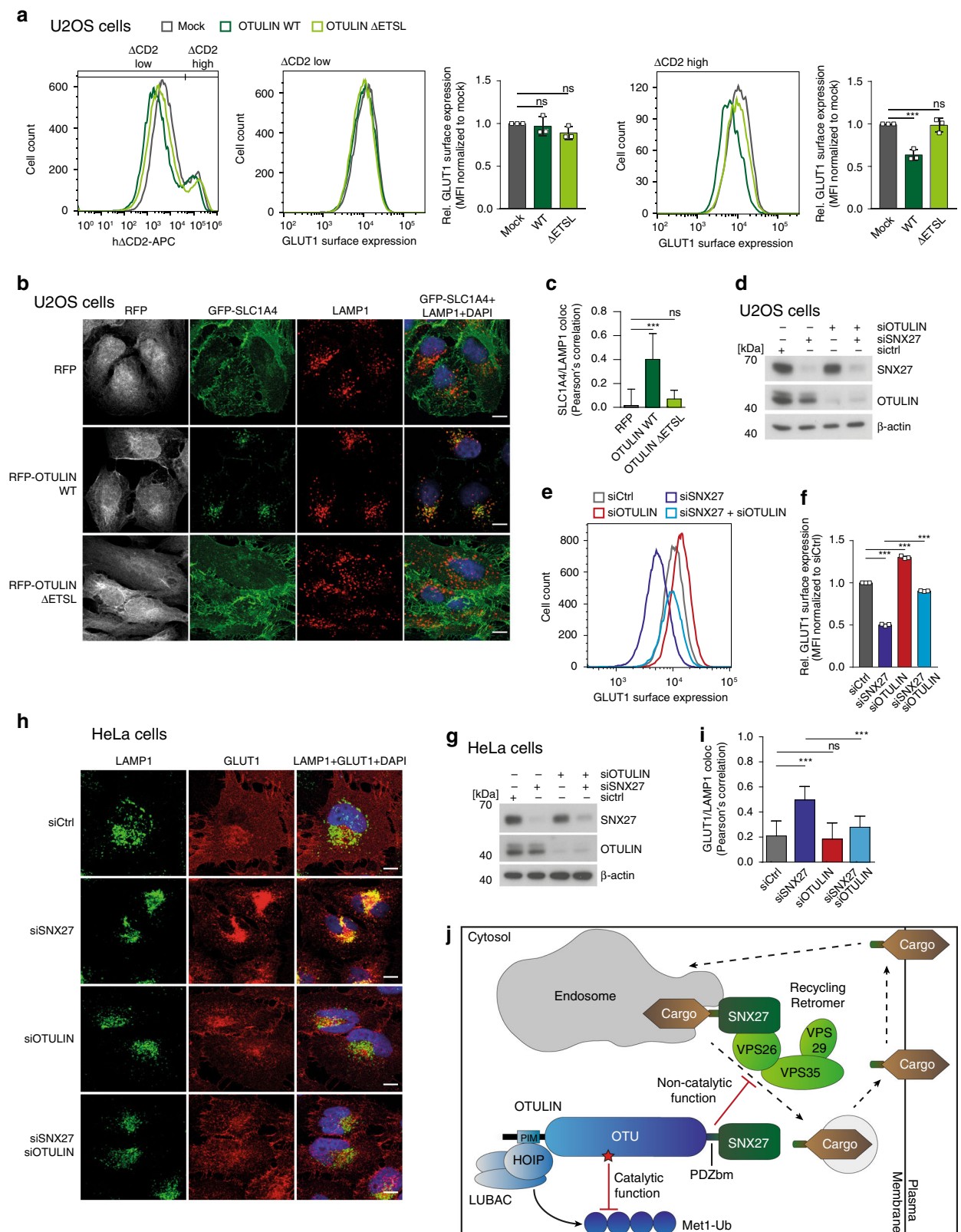

explore this further and to validate the results in an independent setting, we used confocal microscopy to determine the lysosomal localization of GLUT1 after knock-down of SNX27, OTULIN, or SNX27/OTULIN in HeLa cells. As observed earlier, diminished GLUT1 recycling coincided with increased lysosomal accumulation as evidenced by LAMP1 co-localization in SNX27 knock-down cells compared to siCtrl-treated cells (Fig. 8g–i). While siOTULIN did not affect lysosomal sorting of GLUT1, depletion of OTULIN in SNX27 knock-down cells prevented increased GLUT1 sorting to lysosomal vesicles caused by the SNX27 single knock-down. Taken together, the results demonstrate that by antagonizing cargo loading OTULIN raises the threshold for

**Fig. 8** OTULIN binding to SNX27 limits recycling of GLUT1 and SLC1A4 to the cell surface. **a** GLUT1 surface staining in mock, OTULIN WT and OTULIN ΔETSL transfected U2OS cells was performed using GLUT1.RBD.GFP reagent on living cells. GLUT1 surface expression was analyzed in untransfected (ΔCD2-low) and in transfected (ΔCD2-high) cells by FACS (left). Relative GLUT1 surface expression was determined as depicted in the histogram and changes in median fluorescence intensity (MFI) were normalized to mock. Graphs represent the mean ± SD of three independent experiments. Two-tailed $p$-values: ns not significant, ***$p \leq 0.001$ by unpaired $t$-test. **b** U2OS cells lentivirally transduced with GFP-SLC1A4 (green) together with RFP, RFP-OTULIN, or RFP-OTULIN ΔETSL (gray) were stained for endogenous LAMP1 (red) and analyzed for SLC1A4/LAMP1 co-localization by confocal microscopy. Scale bars: 10 μm. **c** Co-localization was quantified by determination of Pearson's correlation analyzing at least 14 random pictures and imaging of more than 100 cells for each condition. Graphs depict the mean ± SD. Two-tailed $p$-values: ns not significant, ***$p \leq 0.001$ by unpaired $t$-test. **d** Knock-down in U2OS cells after siRNA transfection was verified by WB. **e** GLUT1 surface staining in U2OS cells after siRNA transfection as indicates was performed using GLUT1.RBD. GFP reagent on living cells. GLUT1 surface expression was analyzed by FACS. **f** Changes in median fluorescence intensity (MFI) were normalized to siRNA control to calculate relative GLUT1 surface expression. Graphs represent the mean ± SD of three independent experiments. Two-tailed $p$-values: ***$p \leq 0.001$ by unpaired $t$-test. **g** Knock-down in HeLa cells after siRNA transfection was verified by WB. **h** HeLa cells transfected with siRNA as indicated and stained for endogenous GLUT1 (red) and the lysosomal marker LAMP1 (green). Co-localization of GLUT1 and LAMP1 was analyzed by confocal microscopy. Scale bars: 10 μm. **i** Co-localization was quantified by determination of Pearson's correlation analyzing at least eight random pictures and imaging of more than 100 cells for each condition. Graphs depict the mean ± SD. Two-tailed p-values: ns not significant, ***$p \leq 0.001$ by unpaired $t$-test. **j** Schematic model for the dual function of OTULIN. Through LUBAC binding and catalytic activity OTULIN controls Met1-ubiquitin chain homeostasis. By binding to SNX27 OTULIN counteracts cargo recruitment and retromer assembly to antagonize endosome-to-plasma membrane trafficking of internalized cargos. Source data are provided as a Source Data file

efficient endosome-to-plasma membrane recycling by the SNX27-retromer.

## Discussion

By identifying SNX27 as an interactor of OTULIN, we have uncovered an unexpected role of OTULIN in the regulation of endosome-to-plasma membrane recycling (Fig. 8h). Contrary to the well-described function of OTULIN in counteracting Met1-linked ubiquitin chains synthesized by LUBAC, the impact of OTULIN on the SNX27-retromer is independent of DUB activity and is driven by the high affinity binding of OTULIN to the cargo binding pocket in the SNX27 PDZ domain. OTULIN-bound SNX27 is not found at early endosomes, but localizes to the cytosol. In line, OTULIN is not a trafficking cargo, but counteracts SNX27 recruitment to typical cargos and retromer sub-units VPS26/VPS35 thereby preventing assembly of a functional SNX27-retromer complex. In consequence, OTULIN antagonizes SNX27-dependent recycling of the nutrient transporters GLUT1 and SLC1A4 and potentially many cell surface proteins.

Alike typical SNX27 PDZ domain cargos, OTULIN contains a conserved C-terminal class I PDZbm that is essential for SNX27 binding[18]. Already the isolated OTULIN PDZbm peptide exceeds affinity of all known PDZbm peptides by at least a factor of two (e.g. as determined for DGKζ and PLHPP1) and increasing up to a factor of 70 or more as for instance in the case of LRRC3B or β2-AR[17]. Whereas in the structure of SNX27$_{PDZ}$ bound to the DGKζ peptide residues upstream of the P-5 position are not important for PDZ binding[17], Val346 (P-6) and Arg345 (P-7) of OTULIN form hydrogen bonds to the side chain of SNX27, which may explain the high affinity of the OTULIN PDZbm peptide. So far, structural and biophysical data on PDZbm-containing SNX27 cargos have focused on isolated cargo peptides[17,22]. By solving the structure of SNX27$_{PDZ}$ in complex with OTULINcat, we have been able to define a secondary interface outside the canonical class I PDZbm. This interface comprises conserved residues of the OTULIN catalytic domain that are not involved in catalytic activity, but are contacting the SNX27 β3–β4 hairpin loop extension. Compared to the isolated OTULIN PDZ-binding motif, affinity of OTULINcat to SNX27$_{PDZ}$ is increased by >30-fold, yielding a $K_D$ of ~31 nM – to our knowledge the tightest interaction ever measured for a PDZ interactor. Structure-guided mutations in the secondary interface diminish the affinity to the level of the isolated OTULIN PDZbm peptide, underscoring that it is required for tight binding, but not for initial recruitment. Recently, it was found that a secondary

interface also augments the affinity of the PlexinB2 cytoplasmic region and the PDZ of RhoGEF by ~10-fold to a $K_D$ of 2 μM[30]. Even though this binding affinity is still far off the $K_D$-values measured between OTULINcat and SNX27$_{PDZ}$, these data suggest that the existence of secondary interfaces may be a more general concept to modulate affinity, and thus potentially selectivity, of PDZ interactors.

In line with the high affinity in vitro, mutations on both sides of the canonical or non-canonical interfaces abolish or reduce cellular binding of full length OTULIN and SNX27, respectively. The endogenous OTULIN-SNX27 complex was detected by various means, such as OTULIN ABP pull-downs, co-immunoprecipitation and confocal microscopy. Surprisingly, despite the existence of more than 250 PDZ domains in over 150 human proteins[31], two independent mass spectrometry approaches yielded exclusively SNX27 as an OTULIN interactor. This high selectivity was unexpected, because the typical class I PDZbm may potentially associate with multiple PDZ domains. However, the β3–β4 hairpin extension, which forms the surface of the secondary OTULIN binding interface on SNX27, is exclusively found in the PDZ domain of SNX27[22]. Indeed, our modeling demonstrated that other PDZs either fail to facilitate OTULIN binding or even clash with the OTU domain and binding studies confirmed severe reduction (at least 700-fold) or complete absence of OTULIN binding to other PDZ domains. Thus, the architecture of the OTULIN-SNX27 complex and the conformation of the secondary interface not only explains the tight binding, but also the high selectivity of OTULIN for the PDZ domain of SNX27.

Pointing out the cellular role of the OTULIN-SNX27 interaction, the structural overlay of SNX27 bound to OTULIN or VPS26A revealed that the secondary binding site of OTULIN overlaps with the contact points of the SNX27 β3–β4 loop and VPS26A[22]. The overlapping interfaces of OTULIN and VPS26A prevent OTULIN recruitment to the SNX27-VPS26 recycling retromer and thus OTULIN is not a typical SNX27 cargo. Quite contrary, OTULIN has a much higher affinity for SNX27 compared to VPS26A and thus antagonizes VPS26A binding, recruitment of typical cargos and endosomal localization of SNX27. In line, plasma membrane expression of the SNX27-VPS26 cargos GLUT1 and SLC1A4 is controlled by OTULIN. In contrast, binding of Met1-linked ubiquitin to the OTU domain and HOIP binding to the N-terminal PIM of OTULIN is not affected by SNX27. In fact, a ternary HOIP-OTULIN-SNX27 complex can form, but mutually exclusive binding of OTULIN

and VPS26 to SNX27 prevents that such a complex could possibly function as a platform to recruit LUBAC to the retromer. SNX27 was pulled-down with active substrate-bound OTULIN and did not affect OTULIN hydrolase activity. SNX27 deficiency did not change abundance of cellular Met1 ubiquitin chains or LUBAC and no effects on TNF, TCR/CD28 or WNT3A signaling were observed. Thus, even though LUBAC-OTULIN-SNX27 complex can form, there is currently no evidence that SNX27 directly impacts on OTULIN- and LUBAC-dependent ubiquitin homeostasis and signaling.

Comparative analytical size exclusion chromatography from cytosolic extracts of Jurkat T cells revealed that endogenous OTULIN and SNX27 are found in a quite stable complex with an apparent 1:1 stoichiometry, which was also determined in vitro. The OTULIN-SNX27 complex is not bound to the VPS26-VPS35 retromer complex or LUBAC. Given the crucial function of SNX27 in retromer-dependent endosome-to-plasma membrane cargo trafficking, this stable association with OTULIN was unexpected. Even though preparation of extracts for gel filtration may favor cytosolic components over proteins bound to endosomal membranes, the data suggest that only a fraction of SNX27 associates with the retromer. The architecture of the membrane-bound VPS26-VPS35-VPS29 retromer complex has been determined[32], but how the SNX27-retromer is assembled in cells, how SNX27-cargos are selectively loaded on or unloaded from the retromer is largely unknown. In fact, endogenous SNX27-VPS26A retromer complexes have not yet been reported after immunoprecipitation indicating a low abundance or a highly dynamic association. Our data implicate that OTULIN could be involved in regulating the endosomal SNX27-containing retromer, whereby OTULIN sequesters SNX27 in the cytosol and SNX27 must be released to allow retromer assembly. Alternatively, through tight binding and the ability to outcompete retromer and cargo binding on SNX27, OTULIN may facilitate the liberation of SNX27 after cargos have reached their destination.

OTULIN has been identified as part of the SNX27 interactome, but despite the high affinity it was just one of many interactors[18,33]. As a cargo-recruiter, SNX27 is expected to bind to a large interaction network and several mechanisms may account for the broad specificity of SNX27. We know affinities of PDZbm peptides, but native cargos may differ markedly as seen in the case of OTULIN. Further, affinities may change in the context of the endosomal retromer complex. While OTULIN cannot bind VPS26A-bound SNX27, affinities of GLUT1 and Kir3.3 PDZbms to SNX27 in complex with VPS26A are increased by an order of magnitude[22], indicating that upon retromer assembly conformational changes may favor cargo recruitment. In addition, cargo-engagement or release may be modulated by endosomal retromer interactors like ANKRD50, which as a PDZbm-containing protein enhances SNX27-retromer recycling[29]. Finally, PDZbm phosphorylation could act as a switching mechanism for PDZ domain interactions[34]. Interestingly, replacing threonine 350 with a phospho-mimetic aspartate in OTULIN PDZbm (-2 position) reduces the $K_D$ of OTULIN towards SNX27 to the affinity of a prototypic cargo. Thus, phosphorylation in the PDZbm or modifications in the secondary interface of OTULIN may destabilize the OTULIN-SNX27 complex. Vice versa, phosphorylation of some cargo peptides increases their affinity for SNX27, which may further shift the balance from OTULIN to cargo binding[17]. Detailed analyses will be required to assess the impact of conformational changes and post-translational modifications in the regulation of retromer assembly/disassembly. Of note, in size exclusion chromatography absence of OTULIN alone is not sufficient to shift SNX27 to VPS26/VPS35-containing fractions. Most likely the low affinity of

SNX27 and VPS26A explains why the retromer complex is not maintained during chromatography. Besides OTULIN, the phosphatase PTEN was suggested to counteract recycling by hindering SNX27 cargo recruitment and retromer assembly[35]. The C-terminal PDZbm of PTEN associates with SNX27 PDZ with a $K_D$ of 37 μM, which is quite low for a potential competitive interactor. However, the data suggest that other proteins besides OTULIN may control SNX27 functions possibly in either a cargo-specific manner or at different stages of the trafficking process.

OTULIN deficiency or loss of function mutations cause severe phenotypes that so far have been associated with hydrolase activity[13,14]. Human hypomorphic germline mutations that diminish recruitment of Met1-linked ubiquitin to OTULIN (e.g. OTULIN[L272P]) trigger OTULIN-related auto-inflammatory syndrome (ORAS)[13,15]. ORAS phenotypes are mimicked by loss of OTULIN in myeloid cells[13]. The dominant effect of OTULIN catalytic activity precludes resolving the physiological role of the OTULIN-SNX27 complex in available murine models. While SNX27[-/-] mice are not viable[19], SNX27[+/−] heterozygosity leads to neuronal defects[36]. Whereas deficits in synaptic function, learning and memory have been associated with Down syndrome, promotion of β-amyloid generation may enforce Alzheimer pathogenesis[36,37]. Thus, neuronal pathways may be highly sensitive to modulation of SNX27 availability. Specialized mouse models are needed to address the in vivo function of OTULIN-SNX27 interaction and to reveal putative involvements in inflammatory, neuronal or other diseases.

Taken together, our data provide evidence that OTULIN exerts a dual effect and regulates two critical processes required for cellular homeostasis. While OTULIN catalytic function prevents an overload of Met1-linked ubiquitin chains synthesized by LUBAC, OTULIN also regulates SNX27-dependent cargo loading, retromer assembly and endosome-to-plasma membrane recycling via a non-catalytic mechanism.

## Methods

**Cell lines and primary cells**. HEK293 (RRID: CVCL_0045, DSMZ), HEK293T (RRID: CVCL_0063, DSMZ), Hela (RRID: CVCL_0030; DSMZ) and U2OS (RRID: CVCL_0042; ATCC) cells were cultured in DMEM (high glucose) and Jurkat T cells (laboratory of Lienhard Schmitz; verified by DSMZ) in RPMI 1640 supplemented with 10% fetal calf serum (FCS), 100 U/ml penicillin and 100 μg/ml streptomycin. MEFs from WT and OTULIN KO mice were isolated and cultured in DMEM. Primary murine CD4 T cells were isolated from spleen and lymph nodes from C57BL/6 J WT mice by MACS purification using a murine CD4 T cell isolation kit (Miltenyi Biotec). Isolated cells were cultured in primary T cell medium (RPMI medium supplemented with 10% heat-inactivated FCS, 100 U/ml penicillin, 100 μg/ml streptomycin, 1% NEAA, 1% HEPES, 1% L-glutamine, 1% sodium pyruvate and 0.1% β-mercaptoethanol) and pre-stimulated with plate-bound anti-CD3/28 antibodies (BD Pharmingen), for 48 h. For cell expansion recombinant IL-2 (1:5000) was added to the medium.

**DNA plasmids**. DNA constructs for this study were generated by common molecular biological techniques using TOP10 or Stbl3 Chemically Competent E. Coli. cDNAs were cloned into a modified pEF4 (Thermo Scientific), a modified pcDNA3.1 or the pEGFP-C1 backbone (Clontech). Primer sequences are provided in Supplementary Table 2. For lentiviral transduction, cDNAs were cloned into a pHAGE-hΔCD2-T2A, pHAGE-GFP, or pHAGE-RFP vector. Mutations were introduced by site-directed mutagenesis. His6-3C-OTULINcat and His6-3C-SNX27[PDZ] were cloned into the pOPIN-B vector, and PDZ domains of RhoGEF, PSD95, NHERF1, MUPP1, Syntenin1 and LNX2 were cloned from synthesized codon-optimized DNA fragments (GeneArt, Invitrogen; Supplementary Table 3) and cloned into the pOPIN-S vector. PDZ domains of SHANK1-3 were cloned from a human brain cDNA library (Clontech) into the pOPIN-S vector.

**Cell transfection**. For transient overexpression of proteins, HEK293 cells were transfected with the respective plasmids using standard calcium phosphate transfection protocols. HEK293T, U2OS and Hela cells were transfected using X-tremeGENE HP DNA Transfection Reagent (Roche) according to the manufacturer's instructions. For knockdown experiments, cells were transfected with SMARTpool siRNA (50 nM, Dharmacon) and Lipofectamine RNAiMAX reagent

(Thermo Scientific; U2OS) or Atufect transfection reagent (Silence Therapeutics; Hela) and analyzed after 72 h.

**Antibodies**. Following antibodies were used at a dilution of 1:1000 except where otherwise stated: Anti-Actin (sc-1616 HRP; 1:10,000), anti-ATP7A (sc-376467; 1:100), anti-normal mouse IgG (sc-2025; 1.4 μg for IP), anti-ERK (sc-514302) from Santa Cruz; anti-GFP (2555), anti-OTULIN (14127; 1.4 μg for IP), anti-IκBα (4814), anti-phospho-IκBα (Ser32/36) (9246), anti-LAMP1 (15665; 1:100), anti-phospho-ERK (Thr202/Tyr204) (9101), anti-normal rabbit IgG (2729; 1.4 μg for IP), anti-EEA1 (3288, 1:100) all CST; anti-SNX27 (ab77799; 1.4 μg for IP and 1:50 for IF), anti-VPS26 (ab23892), anti-VPS35 (ab157220; 1:10,000), anti-MRP4 (ab15602; 1:2500), anti-DGKζ (ab105195; 1:100), anti-GLUT1 (ab15309; 1:100 and ab115730) all Abcam; anti-Beta PIX (07-1450-I; 1:5000), anti-Met1-Ub (MABS199) all Millipore; anti-STEAP3 (17186-1-AP), anti-KIDINS220 (21856-1-AP) all Proteintech; anti-SNX27 (for murine SNX27) gift from W. Hong; anti-Beta-catenin (610153, 1:2000) BD; anti-FLAG M2 (F3165; 1:10000 and 1 μg for IP) Sigma; anti-HOIP (MAB8039; 1:5000) R&D; anti-HA (Core facility monoclonal antibodies HMGU; 25 μl for IP); anti-CD2-APC (17-0029-41; 1:200) eBioscience.

**Western blotting**. Following separation of proteins by SDS-PAGE, proteins were transferred onto PVDF membranes using an electrophoretic semi-dry blotting system. Membranes were blocked in 5% milk/PBS-T for 1 h at room temperature (RT), followed by incubation with primary antibody (2.5% milk/PBS-T) overnight at 4 °C. The next day, membranes were washed three times with PBS-T before adding HRP-coupled secondary antibodies (1.25% milk/ PBS-T; 1 h, RT). For HRP detection by enhanced chemiluminescence, LumiGlo reagent (CST) was used.

**Protein-protein interaction studies**. For GFP-Traps HEK293 cells were washed with PBS and lysed in 500 μl co-IP buffer (25 mM HEPES pH 7.5, 150 mM NaCl, 0.2% NP-40, 10% glycerol, 1 mM DTT) with phosphatase inhibitors (10 mM sodium fluoride, 8 mM β-glycerophosphate, 300 μM sodium vanadate) and protease inhibitor cocktail (Roche) (20 min, 4 °C) one day after transfection. Lysates were cleared by centrifugation at 20,000 × g (10 min, 4 °C). In total, 30 μl was removed from each lysate and mixed with reducing SDS sample buffer (lysate control). Subsequently, extracts were incubated with GFP-Trap beads (Chromotek), equilibrated in co-IP buffer, for 1–2 h on a rotator at 4 °C. To pull down GFP-tagged proteins, samples were centrifuged at 2500 x g (2 min, 4 °C). Beads were washed three times in co-IP buffer without inhibitors and finally resuspended in 25 μl 2x SDS sample buffer. PD eluates as well as lysate control samples were analyzed by antibody detection on Western Blot membranes after SDS-PAGE.

For the analysis of interactions between SNX27 and cargos, cells were lysed in a buffer without salt (50 mM Tris·pH 7.4, 0.5% NP-40) supplemented with phosphatase and protease inhibitors according to[22]. Cargo binding to SNX27 was quantified using the Fiji/ImageJ software[38]. Following Western Blotting, protein band areas were measured with the Gel Analyzer tool and normalized to precipitated SNX27.

For co-immunoprecipitations (co-IP) of endogenous proteins, $2.5–5 \times 10^7$ cells were lysed in 500 μl co-IP buffer supplemented with phosphatase and protease inhibitors (20 min, 4 °C). For IPs of overexpressed proteins from HEK293 extracts, fewer cells (~$1 \times 10^7$) were lysed one day after transfection. Lysates were centrifuged at 20,000 × g (10 min, 4 °C) and incubated with the respective antibodies on a rotator over night at 4 °C. The next day, 15–25 μl Protein G Sepharose (50% suspension, Thermo Scientific) was added and incubated for another 1–2 h at 4 °C. IPs were washed three times with 1 ml co-IP buffer without inhibitors (700 × g, 2 min, 4 °C) and subsequently denatured by boiling in 25 μl 2x SDS sample buffer. IPs and lysate controls were analyzed by Western Blot after SDS-PAGE.

For the analysis of interactions between endogenous SNX27 and cargos, cells were lysed in a buffer without salt (50 mM Tris·pH 7.4, 0.5% NP-40) supplemented with phosphatase and protease inhibitors according to[22]. Cargo binding to SNX27 was quantified using the Fiji/ImageJ software[38]. Following Western Blotting, protein band areas were measured with the Gel Analyzer tool and normalized to precipitated SNX27.

For OTULIN pull-downs (PD) using the activity-based probe (ABP), $2 \times 10^7$ cells were lysed in 500 μl co-IP buffer with phosphatase inhibitors but without protease inhibitors (20 min, 4 °C). Cell debris was removed from the lysates by centrifugation (20,000 × g, 10 min, 4 °C). To avoid unspecific binding of proteins to the streptavidin beads, lysates were subjected to a pre-clearing with 15 μl High Capacitiy Streptavidin Agarose (50% suspension, Thermo Scientific). After incubation on a rotator (1 h, 4 °C), beads were separated by centrifugation (2500 × g, 2 min, 4 °C). To check whether equal cell numbers were utilized, 30 μl was removed from each sample (lysate) and denatured by boiling in SDS sample buffer for later analysis by Western Blot. Pre-cleared lysates (450 μl) were incubated with 2 μg OTULIN ABP for 15 min at RT, enabling the covalent attachment of active OTULIN to the probe. To bind and pull-down OTULIN-ABP complexes, 25 μl streptavidin agarose (50% suspension) was added to the samples and incubated on a rotator for 2 h at 4 °C. After a first centrifugation step (2500 × g, 4 °C, 2 min), beads were washed three times with 1 ml co-IP buffer without inhibitors. Protein complexes were eluted by boiling beads in 25 μl 2x SDS sample buffer and analyzed by Western Blot after SDS-PAGE, as well as the lysate controls.

For the identification of OTULIN interactors by mass spectrometric analysis of OTULIN ABP-PDs, the individual samples were prepared in quadruplicate. For each sample, $5 \times 10^7$ cells were lysed in 1 ml co-IP buffer without protease inhibitors. Extracts were pre-cleared using 25 μl streptavidin agarose. For ATP-depletion, all pre-cleared lysates (950 μl) were incubated with recombinant apyrase (NEB, 4 U, 30 min, 4 °C). To inhibit OTULIN, samples were incubated for another 30 min at 30 °C with (or without) PR-619 (Merck Chemicals) (250 μM). Extracts were then treated with (or without) OTULIN ABP (5 μg) for 15 min at RT before adding 35 μl streptavidin agarose to pull down OTULIN-ABP complexes and associated proteins (2 h, 4 °C). Beads were washed four times with 1 ml co-IP buffer (2500 × g, 2 min, 4 °C). PDs were eluted by boiling the beads in 50 μl 2x SDS sample buffer. 2.5 μl (5%) was removed for Western Blot analysis, the rest was analyzed by LC-MS/MS.

For the OTULIN interactome determination by LC-MS/MS, Strep-PDs were performed in extracts of OTULIN KO Jurkat Tcells which are reconstituted with either N-terminally StrepII-StrepII-Flag-tagged (SF-)OTULIN WT or mock (SF-tag only). Every sample was prepared in triplicate. For each sample, $2 \times 10^8$ cells were lysed in 1 ml co-IP buffer with phosphatase and protease inhibitors for 1 h at 4 °C. Cell debris was removed from the lysates by centrifugation (20,000 × g, 20 min, 4 °C), supernatants were transferred into new reaction tubes and diluted with 500 μl co-IP buffer. 15 μl was removed from each sample and mixed with SDS sample buffer, to check later by Western Blot if equal cell numbers were used. Strep-Tactin sepharose (IBA) was pre-equilibrated in co-IP buffer and added to each sample (300 μl of 50% slurry/sample). The mixtures were incubated on a rotator for 16 h at 4 °C. Following incubation, beads were washed three times with 1 ml co-IP buffer without inhibitors (250 × g, 2 min, 4 °C) and subsequently boiled in 150 μl 2x reducing SDS sample buffer. 7.5 μl (5%) was removed for Western Blot analysis, the rest was analyzed by LC-MS/MS.

**Protein digest for mass spectrometry analysis**. ABP-PD and Strep-PD eluates were diluted to 175 μl with ultra-pure water and reduced with 5 μl DTT (200 mM in 0.1 M Tris, pH 7.8) for 30 min at RT. Samples were alkylated with 20 μl iodoacetamide (200 mM in 0.1 M Tris, pH 7.8) for 30 min at RT, followed by protein precipitation using a double methanol/chloroform extraction method[39]. Protein samples were treated with 600 μl methanol, 150 μl chloroform and 450 μl water, followed by vigorous vortexing. Samples were centrifuged at 17,000 × g for 3 min and the resultant upper aqueous phase was removed. Proteins were pelleted following addition of 450 μl methanol and centrifugation at 17,000 × g for 6 min. The supernatant was removed, and the extraction process repeated. Following the second extraction process, precipitated proteins were resuspended in 50 μl Urea (6 M) and dilution to <1 M urea with 250 μl ultra-pure water. Protein digestion was carried out with 0.6 μg trypsin (3 μl; 20 μg in 100 μl Trypsin resuspension buffer) at 37 °C overnight. Following digestion, samples were acidified with 1% formic acid and desalted on C18 solid-phase extraction cartridges (SOLA HRP C18 Cartridge), dried and resuspended in 2% acetonitrile, 0.1% formic acid for analysis by LC-MS/MS.

**Liquid chromatography-mass spectrometry/mass spectrometry**. Liquid chromatography-mass spectrometry/mass spectrometry (LC-MS/MS) analysis was performed in biological quadruplicate using a Dionex Ultimate 3000 nano-ultra high pressure reverse phase chromatography coupled on-line to a Q Exactive High Field (HF) mass spectrometer (Thermo Scientific) as described[40]. Samples were separated on an EASY-Spray PepMap RSLC C18 column (500 mm × 75 μm, 2 μm particle size, Thermo Scientific) over a 60 min gradient of 2-35% acetonitrile in 5% DMSO, 0.1% formic acid at 250 nl/min. MS1 scans were acquired at a resolution of 60,000 at 200 $m/z$ and the top 12 most abundant precursor ions were selected for HCD fragmentation.

**Mass spectrometry interactome data analysis**. From raw MS files, searches against the UniProtKB human sequence data base (retrieved 15.10.2014) and label-free quantitation were performed using MaxQuant Software (v1.5.2). Search parameters include carbamidomethyl (C) as a fixed modification, oxidation (M) and deamidation (NQ as variable modifications, maximum 2 missed cleavages, matching between runs, and LFQ quantitation was performed using unique peptides. Label-free interaction data analysis was performed using Perseus (v.1.5.5.3), and volcano plots were generated using a $t$-test with permutation FDR = 0.01 for multiple-test correction and s0 = 2 (or s0 = 0.2) as cut-off parameters.

**Protein expression and purification**. His6-3C-OTULINcat and His6-3C-SNX27$_{PDZ}$ constructs were expressed in Rosetta2 (DE3) pLacI cells (Merck). Cells were grown at 30 °C in 2xTY medium supplemented with 30 μg/ml kanamycin and 34 μg/ml chloramphenicol to an $OD_{600}$ of 0.6-1.0. The culture was cooled to 20 °C prior to overnight induction with 400 μM IPTG. Cells were resuspended and lysed by sonication in lysis buffer (20 mM Tris pH 7.4, 300 mM NaCl, 50 mM imidazole, 2 mM β-mercaptoethanol) supplemented with lysozyme, DNaseI (Sigma), 1 mM PMSF and protease inhibitor cocktail. Both OTULINcat and SNX27$_{PDZ}$ constructs were purified by immobilized metal affinity chromatography by a HisTrap column (GE Healthcare). Loading was performed in lysis buffer and elution was done using lysis buffer with 500 mM imidazole. Pooled fractions were dialysed overnight into either, cation exchange chromatography buffer (20 mM MES pH 6.0, 50 mM NaCl,

4 mM DTT) for SNX27$_{PDZ}$ constructs, or anion exchange buffer (20 mM Tris pH 8.5, 4 mM DTT) for OTULIN constructs. For biophysical and structural analysis the His6-tag was cleaved by overnight incubation with 3C protease during dialysis. SNX27$_{PDZ}$ variants were purified by cation exchange chromatography using a ResourceS column (GE Healthcare) and eluted in buffer containing: 20 mM MES pH 6.0, 1 M NaCl, 4 mM DTT. OTULIN variants were purified by anion exchange chromatography using a ResourceQ column (GE Healthcare) and eluted in buffer containing: 20 mM Tris pH 8.5, 1 M NaCl, 4 mM DTT. Eluted fractions of SNX27 and OTULIN variants were subjected to size exclusion chromatography (HiLoad 16/60 Superdex 75, GE Healthcare) in buffer containing: 20 mM Tris pH 7.4, 200 mM NaCl, and 4 mM DTT. The resultant fractions were judged to be 95–99% pure following SDS-PAGE analysis and flash frozen. Expression and purification of the aforementioned PDZ domains was analogous to SNX27PDZ with the exception that SUMO protease was added to cleave the His6-SUMO tag.

**Crystallization data collection and refinement**. OTULINcat C129A (150 μM) and SNX27$_{PDZ}$ (167 μM) at a total concentration of 7 mg/ml were grown by sitting-drop vapor diffusion with 1.11 molar excess of SNX27$_{PDZ}$. Large thin plates appeared overnight from reservoir containing: 19% (w/v) PEG 3350, 200 mM Li2 (SO4), 100 mM Bis-Tris pH 6.8 and reached full-size within one week. Crystals were transferred into a solution containing 25% (v/v) glycerol prior to cryo-cooling.

Diffraction data were collected at Diamond Light source beamline IO3. Diffraction images were processed using xia2[41] and manually scaled using AIMLESS[42]. The structure of OTULINcat C129A bound to SNX27$_{PDZ}$ was determined by molecular replacement using PHASER[43] placing SNX27$_{PDZ}$ (PDB ID: 4P2A) and OTULINcat (PDB ID: 3ZNZ). Iterative rounds of model building and refinement were performed with COOT[44] and PHENIX[45], respectively. Data collection and refinement statistics can be found in Table 1. All structure figures were generated with Pymol (www.pymol.org).

**Isothermal calorimetry**. Prior to isothermal calorimetry (ITC) experiments all samples were exchanged into ITC buffer: 20 mM HEPES pH 7.4, 200 mM NaCl, and 2 mM TCEP. ITC data were performed using an Auto iTC200 instrument (Malvern Instruments, Malvern, UK) at 25 °C with the exception of SHANK1-3 PDZ experiments that were performed at 30 °C. Between 20 and 30 μM of OTU-LINcat variant in the calorimeter cell were titrated against 150–450 μM PDZ variant in the syringe using 19 injections of 2 μL preceded by a small 0.5 μl pre-injection. The changes in heat release were integrated over the entire titration and fitted to a single-site binding model using the PEAQ Analysis software (Malvern Instruments).

**Analytical size exclusion chromatography**. Binding studies with purified recombinant SNX27PDZ, OTULINcat, VPS26A and associated complexes were performed on an ÄKTA Pure (GE Healthcare) using a Superdex 75 10/300 column equilibrated in analytical size exclusion chromatography (SEC) buffer (20 mM Tris pH 7.5, 150 mM NaCl, 2 mM DTT). A total of 500 μl of 65 μM complexes at equimolar ratio or in isolation were loaded onto the column. Fractions containing protein were mixed with SDS loading buffer prior to SDS-PAGE analysis.

SEC analyses of Jurkat T cell cell extracts were performed on an ÄKTApurifier System (GE Healthcare). $1 \times 10^8$ Jurkat T cells were washed with PBS and lysed in 400 μl lysis buffer (25 mM HEPES pH 7.5, 150 mM NaCl, 0.2% NP-40, 1 mM DTT) with phosphatase inhibitors (10 mM sodium fluoride, 8 mM β-glycerophosphate, 300 μM sodium vanadate) and protease inhibitor cocktail for 30 min at 4 °C. Lysates were initially centrifuged at $20,000 \times g$ (20 min at 4 °C) and subsequently subjected to an ultracentrifugation step ($125,000 \times g$, 30 min, 4 °C). The resulting supernatants were filtered through Proteus Mini Clarification Spin Columns (Generon; pore size: 0.2 μm). Extracts were loaded onto a Superdex 200 10/300 GL column equilibrated in SEC buffer. In all, 0.5 ml fractions were collected and mixed with SDS sample buffer prior to SDS-PAGE and Western Blot analysis.

**Generation of knock-out cells using CRISPR/Cas9 technology**. The pSpCas9 (BB)-2A-GFP (PX458) vector was manipulated by standard cloning techniques in order to express *S. pyogenes* Cas9 along with GFP and sgRNAs targeting exon 2 of the *OTULIN* gene (5′-ATACATGAAAGAGGGGCATC-3′), exon 1 of the *RNF31/HOIP* gene (5′-GAGAGCTGGCTAGTAGCGGC-3′) or exon 1 of the *SNX27* gene (5′-GGCTACGGCTTCAACGTGCG-3′). In all, $8 \times 10^6$ Jurkat T cells were mixed with 5 μg plasmid in a 4 mm electroporation cuvette and electroporated with a Gene Pulser X System (Bio-Rad; 220 V, 1000 μF). After 1–3 days, GFP-positive cells were sorted using a MoFlow Cytometer (Cytomation) and seeded in 96-well plates at very low density (0.5–5 cells/well) to obtain clonal cell populations. After expansion, cell clones were screened for loss of the respective protein by western blot. In addition, regions of genomic DNA around the targeted sites were amplified by PCR and sequenced in order to verify the KOs on genomic level.

For the generation of HEK293 KO cells, cells were transfected with the respective PX458 plasmids using X-tremeGENE HP DNA Transfection Reagent (Roche). One day after transfection, GFP-positive cells were sorted and also seeded at very low density.

**Lentiviral transduction of cell lines**. For virus production, $1.5 \times 10^6$ HEK293T cells were seeded in a 10 cm dish in 8 ml DMEM. The day after, cells were transfected with 1.5 μg of the packaging vector psPAX2, 1 μg of the lentiviral envelope plasmid pMD2.G and 2 μg of the respective pHAGE-hΔCD2-T2A, pHAGE-RFP, or pHAGE-GFP. In all, 72 h after transfection, the virus-containing supernatants were collected, sterile filtered and added to the cells to be infected, together with 8 μg/ml polybrene. The next day, virus was displaced by culture medium. Following a recovery phase, transduction efficiency was validated by flow cytometry and Western Blot analysis.

**Stimulation of cells**. For stimulation experiments, $3 \times 10^6$ Jurkat T cells were treated with 20 ng/ml TNFα (biomol) or 20 ng/ml Wnt3a (R&D) for the indicated times. For CD3/CD28 co-ligation, cells were stimulated with 0.3 μg anti-CD3 and 1 μg anti-CD28 antibodies in the presence of 0.5 μg rat anti-mouse IgG1 and IgG2a (all from BD Pharmingen).

**Cell surface expression of Glucose transporter 1**. In total, 100,000 cells transiently transfected with expression vectors or siRNAs were collected after 48–72 h and 72 h, respectively, resuspended in 95 μl Buffer A (complete culture medium, 0.1% NaN$_3$ and 1 mM EDTA) and incubated with GLUT1.RBD.GFP (Metafora Biosystems) according to the manufacturer's instructions (20 min, 37 °C). Following incubation, cells were washed three times with Buffer B (PBS, 0.1% NaN$_3$, 1 mM EDTA and 2% FCS) and analyzed for GFP-labeled GLUT1 on the cell surface by FACS using an Attune Acoustic Focusing Cytometer (Thermo Fisher Scientific). Cells transfected with pHAGE-hΔCD2-T2A constructs (mock, OTULIN WT, OTULIN ΔETSL) were stained in parallel with anti-CD2 antibody (1:400), in order to distinguish between transfected and untransfected cells by means of the co-expressed cell surface marker ΔCD2.

**Confocal microscopy**. Cells were seeded at low density in 24-well plates on round coverslips coated with Poly-L-lysine solution (Sigma). For staining, cells were fixed for 10 min in 4% PFA (at RT) and subsequently washed thoroughly with PBS. Cells were blocked in blocking solution (PBS, 0.1% Tween-20, 5% BSA) for 1 h at RT, followed by incubation with primary antibody in blocking solution (1:100, 1 h at RT). Then, cells were washed three times for 5 min with PBS-T, incubated in PBS-T with secondary antibodies (all Thermo Scientific) goat α-mouse Alexa Fluor 488 (#A-11001), goat α-mouse Alexa Fluor 647 (#A-21235), goat α-rabbit Alexa Fluor 568 (#A-11011), goat α-rabbit Alexa Fluor 647 (#A-21244) at 1:1000 dilution for 1 h at RT and washed three times with PBS-T. After nuclear counterstaining with DAPI in PBS-T, coverslips were washed in PBS and mounted on glass slides using mounting medium (2.5% 1,4-Diazabicyclo-octane, 10% Mowiol 4-88, 25% glycerol and 0.1 M Tris-HCl). Images were acquired using a Leica SP5 confocal laser scanning microscope (×63 magnification). Images were analyzed using the Fiji/ImageJ software[38]. The Pearson's correlation between the respective channels was quantified with the Coloc2 tool of the Fiji software using the auto-threshold function.

**Reporting summary**. Further information on research design is available in the Nature Research Reporting Summary linked to this article.

## Data availability
The mass spectrometry proteomics data have been deposited to the ProteomeXchange Consortium via the PRIDE[46] partner repository with the dataset identifier: PXD012082. Coordinates for the OTULIN–SNX27 structure have been deposited in the PDB with accession number 6SAK [https://www.rcsb.org/structure/6SAK]. Source data underlying Figs. 1, 2, and 4–8 and Supplementary Figs. 1, 2, and 5–8 are provided as a Source Data file with the paper. All other data are available from the corresponding authors upon reasonable requests.

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

## Acknowledgements

We thank Peter Cullen for providing GFP-SNX27 construct and Wanjin Hong for an antibody recognizing murine SNX27. We thank Verena Dubb for help in generating OTULIN and HOIP KO cell lines. We thank Roman Fischer from the Discovery Proteomics Facility (part of the Target Discovery Mass Spectrometry Laboratory) for his expert help with the mass spectrometry analysis. Access to DLS was supported in part by the EU FP7 infrastructure grant BIOSTRUCT-X (283570). Work in D.Ko. laboratory was supported by the Medical Research Council (U105192732), the European Research Council (724804), and the Lister Institute for Preventive Medicine. Work of P.R.E was funded by the Medical Research Council (MR/R008582/1).

## Author contributions

A.S. conceived and performed cellular experiments, analyzed, and interpreted the data and helped to write the manuscript. P.R.E. and D.Ko. conceived and performed structural and biophysical studies, and helped to write the manuscript. A.P.F, S.B., and B.M.K. conceived, performed, and analyzed mass spectrometry experiments with material synthesized and provided by F.E.O.; A.Sc., L.H., and P.T.P. helped with acquisition and analyses of microscopy data. K.Ku. and K.Ke. helped with generation of tools and carrying out critical experiments and binding studies. D.K. conceived the study and experiments, wrote the manuscript, and secured funding. All authors read, acknowledged, and helped with the final version of the manuscript.

## Additional information

Competing interests: F.E.O is the co-founder and shareholder of UbiQ Bio BV. The remaining authors declare no competing interests.

