## [Peer Review File · Nature Communications]

Reviewers' comments:

Reviewer #1 (Remarks to the Author):

In this manuscript Weber et al. reveals a novel and unexpected role of the deubiquitinase OTULIN related to the regulation of the SNX27-retromer dynamics and cell surface cargo recycling. This function, the authors claim, is not dependent on either the deubiquitinase catalytic activity of OTULIN or on the LUBAC homeostasis, defining a new moonlighting function for OTULIN. The manuscript displays compelling data on the interaction of OTULIN with the PDZ domain of SNX27 by MS and pull-down analyses in cultured cells. A crystal structure of the complex between OTULIN-SNX27 also reveals the atomic details of the interaction of OTULIN with the SNX27PDZ domain. Most of the structural details disclosed by the crystal structure are nicely checked by mutagenesis using ITC and pull-down experiments in cultured cells. Finally, the authors demonstrate the role of OTULIN in the SNX27-cargo dependent sorting pathway, but its real relevance of this mechanism in the cell is perhaps the only point that needs more solid data. Globally, the manuscript is very well structured, the objectives are correctly proposed and developed with high quality experiments, and the flow of the results runs nicely throughout all the manuscript. In summary, Weber's et al. manuscript reveals a novel and unexpected role of OTULIN regulating endosomal-sorting processes by a strong interaction with SNX27, and I believe that the novelty and quality of the results justify its publication.

Just a few comments on the results,

1. A major question that comes to my mind after reading the manuscript is to figure out whether this novel activity of OTULIN in the regulation of the SNX27 binding to the VPS26-35-29 retromer really happens in the cells. The last experiment of the GLUT1 substrate expression in the U2OS cells is quite compelling (fig 7f). However, perhaps it would be a good idea to perform similar experiments knocking out endogenous OTULIN and to figure out the levels of GLUT1 or other sorting-related substrates (perhaps using cultured cells from OTULIN knockout mice ??). This experiment would strengthen the biological role of OTULIN in protein sorting, and would prove that OTULIN and SNX27 are participating indeed in the same cellular pathway.
2. In page 8, the manuscript describes the stabilization of alpha helix ($\alpha 2$), by the formation of a Cys347 N-terminal cap. However I could not find the numbering of the secondary structure of OTULIN in any figure (from the figure it is hard to see where this helix is located). Perhaps labeling fig 3a is enough.
3. Since loop b3-b4 is so important in the secondary binding region, and seems to be unique in the SNX27 PDZ domain, it would be interesting to include a sequence alignment (based on the structure) of the b3-b4 region in other PDZ domains (at least with the PDZ domains used in figure 5).
4. Many experiments of the manuscript rely on mutagenesis of either OTULIN or SNX27 PDZ domain. Despite that ITC results are quite convincing and make sense with the whole story, it would be nice to check the quality of the point mutants. For example with OTULIN point mutants, since it is a deubiquitinase, it would be relatively easy to conduct some deubiquitinating activities comparing WT and mutants.
5. I think that Figure 4b should be better displayed in stereo.
6. At the end of page 13, it is strange that SNX27 is not shifted to the VPS26-35 retromer fractions in OTULIN KO gel filtration analysis. Perhaps this is because of the weak interaction between SNX27 and VPS26-35 retromer ???. The authors should at least discuss this point.
7. Since the authors claim that there is a competition in the binding between SNX27 and OTULIN or VPS26-35 retromer, in light of figure 5, it would be interesting to conduct some competition binding analysis in vitro. I think that this experiment might be quite informative to support the major hypothesis of this work on the control of cargo binding by OTULIN. There should exist a competition between OTULIN and VPS26-35 to bind SNX27.

8. Fig.7c is quite relevant to show the decrease in the SNX27-cargo binding by OTULIN expression, it would be important to show some statistics based from the WB analysis.

9. In fig 7b I don't observe the GFP-Trap precipitated retromer subunit VPS26 and VPS35, as stated in page 15. Please confirm or correct.

Reviewer #2 (Remarks to the Author):

In the manuscript entitled 'Regulation of the endosomal SNX27-retromer by OTULIN', Weber and her colleagues have identified a new interacting partner of OTULIN by mass spectrometry. The authors used the biotin-labelled OTULIN activity-based probe (ABP), which made it success to identify new interacting proteins including SNX27. Interaction of endogenous proteins of OTULIN and SNX27 was confirmed in lysates derived from different cell types. The interaction is through the SNX27-PDZ and the c-terminal motif of OTULIN, mediated by the second interface giving specificity. Biologically, this interaction does not affect OTULIN deubiquitinase activity and does not affect LUBAC-dependent cellular signalling. SNX27 is known to interact with VPS26A to form a retromer complex and interaction with OTULIN is mutually exclusive, thus antagonizing the endosomal SNX27-VPS26/VPS35 retromer complexes in cells. The model cargo glucose transporter 1 (GLUT1) loading to SNX27 PDZ was impaired in the presence of OTULIN. In this way, the authors demonstrated for the first time that OTULIN was shown to be involved in cell trafficking and endosome-to-plasma membrane recycling. The manuscript is very well written and the biochemical properties of OTULIN -SNX27 interaction are shown very clearly and in a convincing manner. Most importantly, this is totally a novel function of OTULIN. The biological implication in the regulation of endosomal SNX27 retromer by OTULIN is very interesting yet this part requires more investigation.

Major points:

1. Figure 7f. It is important to examine the GLUT1 surface expression level in different cell types not only in U2OS. Moreover, for making a conclusion of OTULIN functions in the regulation of the retromer complex, it is important to investigate different target surface proteins, at least one more, regulated by the OTULIN-SNX27 complex, possibly by using a similar approach as used in Fig 7f.

Minor points:

1. Supp Fig 4. It is critical to examine TNF-induced phosphorylation and degradation of I κ B α in the OTULIN KO clones as a control.
2. Having a schematic of the OTULIN function in the regulation of the retromer complex is suggested to help readers.
3. in page 5 in the last paragraph; 'To prove that SNX27 is directly associating with OTULIN in cells'.... The author's approach can show that they make a complex in cells but cannot show 'direct' association. Thus, it is suggested to rephrase this part.

Reviewer #3 (Remarks to the Author):

The authors have identified a PDZ-PDZbm molecular interaction between SNX27 and OTULIN which appears to be stabilized by a second interface. The biochemical/structural identification of this interaction is complete. However, the context of this interaction within the cell is unclear and needs to be further investigated. Most of the evidence presented is consistent with this interaction occurring in the cytoplasm rather than on membranes. This needs to be directly examined. The conclusions about the relevance of this molecular interaction to the functional action of SNX27 is weak and maybe indirect. A real option is that high levels of expression of OTULIN sequesters SNX27 through a molecular interaction in the cytoplasm. The evidence that this interaction is actually of physiological relevance is indirect and evidence supporting it needs to be improved.

Specific Points to be addressed

- 1) The molecular interaction work is messy in that authors nominate to use a range of engineered or endogenous OTULIN constructs across the different experiments. These experiments need to be repeated using a consistent bait for the different co-ip experiments.
- 2) Retromer will associate with SNX27 on endosome membranes in the presence of cargo. The current working model supported by the biochemistry is that retromer will not associate with SNX27 in the cytosol. As OTULIN-SNX27 interaction appears to be in the cytosol it will not be competing with OTULIN.
- 3) Direct evaluation of the impact OTULIN has on the membrane recruitment of SNX27 needs to be examined.
- 4) Figure 6c magnification details required.
- 5) Figure 6c data is poor and potentially selective. The level of expression GFP-SNX27 is lower in the endosome associated images compared to the others. More representative images need to be presented. In fact it would be best just to examine the impact of endogenous SNX27 in these experiments.
- 6) The deltaETSL appears to be within the nucleus of the cell shown.
- 7) Do not switch cell lines between the co-ip (6a,b – HEK293) and IF 6c U206 cells. Repeat the experiments throughout this study within the same cell line – this will allow comparison of the results independent of the numerous differences between the cell types.
- 8) “endosomal association of GFP-SNX27 was lost” is not supported by the data presented. You would not observe the endosome associated SNX27 in the images shown. Co-localisation studies need to be presented with endogenous SNX27 to make this conclusion. Inclusion of membrane fractionation would confirm these observations.
- 9) The level of over-expression of OTULIN needs to be documented. A trivial explanation of these results is that non-physiological levels through extreme over-expression may simply saturate the system and just sequester SNX27 (dominant interfering). This is fine to disrupt the system but is not good evidence that the molecular interaction is actually physiologically relevant.
- 10) Is the function of SNX27 impacted in the OTULIN KO cell lines you have generated?
- 11) The data in 6e was concluded that the “majority of OTULIN eluted together with SNX27”. However, SNX27 is predominantly on membrane associated on endosomes by IF shown. Is OTULIN also predominantly on endosomes. The inconsistency of these observations needs to be discussed.
- 12) Western blotting/co-immunoprecipitation data should be quantified incorporating data from at least 3 independent experiments. This is highlighted in Figure 7 in which the level of co-ip between the otulin ko (d) and mock (e) which should be the same.
- 13) 7f – how was surface GFP differentiated from intracellular GFP by FACS?
- 14) A more detail investigation of the OTULIN KO cell line needs to be performed to determine if this impacts on SNX27 function. This should examine the relative distribution of SNX27 (does it impact on recruitment to retromer positive endosomes); SNX27 cargo (reduced retrieval to plasma membrane ;higher endosome associated pool of molecules – likely higher rate of degradation). These phenotypes are then rescued by expression of OTULIN (at endogenous levels) but not OTULIN deltaETSL.

Reviewer #4 (Remarks to the Author):

In the present study, the authors identify a novel interaction between the ubiquitin modifying enzyme Otulin and the retromer component SNX27. Otulin is shown to bind to SNX27 in a non canonical way as it interacts in a bivalent way with this sorting nexin. The PDZ ligand engages the PDZ binding pocket while a second interface contacts SNX27 at a site that partially overlaps with the VPS26 binding site that connects SNX27 to retromer. Because of this mode of interaction, Otulin is demonstrated to have much higher affinity for SNX27 than “regular” PDZ ligands. Since Otulin can displace SNX27 from its cargoes, it is potentially a novel regulator of the SNX27-retromer.

Technically, the study is of very high quality. The novel interaction is clearly delineated and

supported by extensive crystallography and convincing measurements of the respective affinities. Otulin is surely a high affinity interactor of SNX27. That said, some doubts remain about the functional part of this study and I am not completely convinced that Otulin is in fact a regulator of the retromer pathway. SNX27 interacts with a wide range of PDZ ligands, most of them located at the C-terminus of integral membrane proteins. There are, however, many intracellular, cytosolic proteins that were shown to engage the SNX27 PDZ domain such as DGKz, ZO2, ANKRD50 and many others. Steinberg et al. (2013, Nature Cell Biology, supplementary table 1) have performed quantitative proteomics on GFP-SNX27. While Otulin (FAM105b) was readily detected, it is just one of many PDZ bearing proteins and the proteomics data (score, PSMs) does not indicate a particularly high stoichiometry. Furthermore, Kvainickas et al. (2017, Journal of Cell Science) showed that overexpression of the PDZ ligand bearing SNX27 interactor ANKRD50 has exactly the same effect on SNX27 cargo as Otulin: The ANKRD50 tail displaces GLUT1 (and likely other cargo) just as Otulin does and removal of the PDZ ligand from ANKRD50 suppresses this effect. This indicates that the displacement of cargo through overexpression of SNX27 interactors may not be a unique feature of Otulin. One could argue that Otulin is simply a high affinity interactor for SNX27 of unknown function that can displace other PDZ ligand bearing proteins from SNX27.

To address these concerns, the authors could demonstrate that the effect of Otulin knockout on SNX27 cargo binding is unique. For that they could use CRISPR or siRNA pools (Dharmacon smartpools, for example) to knockout/knockdown proteins such as ANKRD50, ZO2, DGKzeta, ABRO1 and analyze SNX27 cargo binding. Similarly, it would be nice if some of these proteins could be overexpressed to check whether this results in a similar loss of SNX27-cargo interaction as overexpression of Otulin. If they did, this would argue that overexpression of SNX27-PDZ interactors generally perturbs cargo binding. Clearly, the knockdown/knockout data is more relevant here, as the overexpression results may be meaningless.

Additional points:

The authors claim that Otulin displaces SNX27 from endosomes, probably by outcompeting binding to VPS26. They also state that only a fraction of SNX27 is retromer associated, while a significant fraction of SNX27 co-migrates with Otulin. In my experience working with SNX27, both in fixed cells and in living cells, SNX27 is very crisply endosomal with minimal cytosolic background signal. This is odd if Otulin really keeps the majority of SNX27 in a cytosolic complex. Does SNX27 recruit Otulin onto endosomes?

Figure 6C: The colocalization data is not very convincing. The authors should moderately overexpress SNX27 and Otulin and analyze the subcellular distribution of SNX27 using a good confocal microscope and high zoom. GFP-SNX27 tends to be quite cytosolic when overexpressed too much so this needs to be done carefully. Ideally, low level lentiviral expression of SNX27 is used as this results in SNX27 that is properly localized to endosomes.

Figure 7B: The authors state that GFP-SNX27 precipitates VPS26 and VPS35. While this is accurate, it is not shown in the panel.

Figure 7D: The authors need to show the levels of the respective cargoes in the Otulin KO cells. As it is, it is unclear whether the increase in SNX27 cargo binding is really caused by increased binding or whether there is simply more of the respective cargoes in the KO clone.

GLUT1 data: The authors show that overexpression of Otulin reduces surface expression of GLUT1. It would be great if the authors knocked down Otulin and analyzed GLUT1 surface levels following the knockdown. If surface levels were increased, this would show that loss of Otulin promotes the activity of SNX27. The same could be done for SLC1A4, which is another SNX27 dependent surface channel (Kvainickas et al., 2017, Journal of Cell Science).

Minor points:

Page 5: The authors state that they tested whether OTULIN directly binds to SNX27. For that they perform the biotin probe IP in OTULIN KO cells, in which no SNX27 was precipitated. While this is a nice experiment, it does not prove that SNX27 directly interacts with OTULIN. This should be rephrased. Clearly, OTULIN is required to pull down SNX27 with their probe but this is not real

evidence for direct binding as there may still be a third bridging molecule. The same also applies to the 2xStrep-Flag precipitation. The authors would need to purify the PDZ domain and OTULIN in bacteria and test for direct interaction to really prove this. Given that they have a crystal structure of the complex, this seems rather pointless so the authors could simply rephrase these sentences.

For all reviewers:

Please find attached our responses to the reviewer's comments for our manuscript entitled '*Regulation of endosomal SNX27-retromer by OTULIN*', in which we identified and characterized a novel and unexpected cross-talk of the ubiquitin and retromer systems. We feel that we got some very valuable and constructive comments that have helped to improve the study. Wherever possible and reasonable, we have done what we were asked for, as you will see in the specific replies below.

Reviewer#3 and #4 felt that improved imaging data should be added to conclude that OTULIN has an impact on endosomal localization of SNX27. This part in **Figure 6** has been completely revised and extended. Image quality has been improved and by using stable and homogenous lentiviral expression systems for GFP-SNX27 and RFP-OTULIN we avoid monitoring overexpression in only a few cells (**new Figure 6b, c**). Further, we demonstrate and quantify reduced co-localization of endogenous SNX27 with EEA1 upon RFP-OTULIN transduction (**new Figure 6d**) and mildly augmented SNX27/EEA1 co-localization in OTULIN KO cells (**new Figure 6e**).

All reviewers recommended to strengthen the part on the biological relevance of the OTULIN-SNX27 interaction for endosomal sorting processes and we have performed several experiments to address the questions in a **new Figure 8**. We demonstrate in U2OS and HeLa cells that OTULIN, but not SNX27-binding deficient OTULIN Δ ETSL, impairs GLUT1 cell surface expression (**Figure 8a, new Supplementary Figure 8a**). To provide evidence that this is not just a GLUT1-specific effect, we validated defective SNX27-retromer trafficking for a second cargo. We used the SNX27 sorting reporter GFP-SLC1A4 (an amino acid transporter), which we stably transduced into U2OS and HeLa cells. By confocal microscopy we demonstrate that GFP-SLC1A4 accumulates in LAMP1-positive lysosomes upon lentiviral expression of RFP-OTULIN WT, but not Δ ETSL (**new Figure 8b,c, Supplementary Figure 8b, c**). We used siRNA knock-down (KD) to analyze, if downregulation of endogenous OTULIN affects cell surface expression of GLUT1, and if OTULIN and SNX27 exert opposing effects in the same cellular recycling pathway. First, we determined GLUT1 on the cell surface by FACS after single or combined KD of SNX27 and OTULIN in U2OS cells (**new Figure 8d-f**). While KD of SNX27 decreased GLUT1 amounts on the cell surface (validating the assay), OTULIN depletion mildly increased GLUT1 at the plasma membrane. Importantly, combined OTULIN/SNX27 KD restored GLUT1 surface expression compared to SNX27 single KD cells. Second, we determined mis-sorting of GLUT1 and co-localization with lysosomes after single or combined KD of SNX27 and OTULIN in HeLa cells (**new Figure 8g-h**). Indeed, while siSNX27 increased GLUT1 on LAMP1-positive lysosomes, siOTULIN alone did not affect lysosomal sorting of GLUT1. However, depletion of OTULIN in SNX27 KD cells prevented increased GLUT1 sorting to lysosomal vesicles caused by the SNX27 single KD. These data provide direct and quantifiable evidence that OTULIN negatively impacts on SNX27-dependent cargo trafficking.

Taken together, we present amazing new data that strengthen our finding on the cellular function of the OTULIN-SNX27 association and we demonstrate that SNX27 and OTULIN participate in the same pathway to regulate endosome-to-plasma membrane trafficking.

Reviewers' comments:

Reviewer #1 (Remarks to the Author):

In this manuscript Weber et al. reveals a novel and unexpected role of the deubiquitinase OTULIN related to the regulation of the SNX27-retromer dynamics and cell surface cargo recycling. This function, the authors claim, is not dependent on either the deubiquitinase catalytic activity of OTULIN or on the LUBAC homeostasis, defining a new moonlighting function for OTULIN. The manuscript displays compelling data on the interaction of OTULIN with the PDZ domain of SNX27 by MS and pull-down analyses in cultured cells. A crystal structure of the complex between OTULIN-SNX27 also reveals the atomic details of the interaction of OTULIN with the SNX27PDZ domain. Most of the structural details disclosed by the crystal structure are nicely checked by mutagenesis using ITC and pull-down experiments in cultured cells. Finally, the authors demonstrate the role of OTULIN in the SNX27-cargo dependent sorting pathway, but its real relevance of this mechanism in the cell is perhaps the only point that needs more solid data. Globally, the manuscript is very well structured, the objectives are correctly proposed and developed with high quality experiments, and the flow of the results runs nicely throughout all the manuscript. In summary, Weber's et al. manuscript reveals a novel and unexpected role of OTULIN regulating endosomal-sorting processes by a strong interaction with SNX27, and I believe that the novelty and quality of the results justify its publication.

Just a few comments on the results,

1. A major question that comes to my mind after reading the manuscript is to figure out whether this novel activity of OTULIN in the regulation of the SNX27 binding to the VPS26-35-29 retromer really happens in the cells. The last experiment of the GLUT1 substrate expression in the U2OS cells is quite compelling (fig 7f). However, perhaps it would be a good idea to perform similar experiments knocking out endogenous OTULIN and to figure out the levels of GLUT1 or other sorting-related substrates (perhaps using cultured cells from OTULIN knockout mice ??). This experiment would strengthen the biological role of OTULIN in protein sorting, and would prove that OTULIN and SNX27 are participating indeed in the same cellular pathway.

As outlined above, we have performed several experiments that strengthen the biological relevance of our finding. Especially, as suggested by the reviewer, we have performed experiments to deplete OTULIN and/or SNX27. However, we used KD instead of KO cells, because many plasma membrane receptors/transporters are essential for cell viability. Thus, clonal selection and adaptation of KO cells may perturb the balance of fast recycling and other transport processes. Further, siRNA KD allows the combined down-regulation of OTULIN and SNX27.

The new data on GLUT1 surface expression and on lysosomal sorting in cells with SNX27, OTULIN or combined SNX27/OTULIN KD strongly support our assumption that the proteins exert opposing effects in the same cellular recycling pathway (**new Figure 8d-i**; please see also above).

2. In page 8, the manuscript describes the stabilization of alpha helix ($\alpha 2$), by the formation of a Cys347 N-terminal cap. However I could not find the numbering of the secondary structure of OTULIN in any figure (from the figure it is hard to see where this helix is located). Perhaps labeling fig 3a is enough.

We have added labels for the alpha helix $\alpha 2$ in **Figure 3A** and **Figure 3C**.

3. Since loop b3-b4 is so important in the secondary binding region, and seems to be unique in the SNX27 PDZ domain, it would be interesting to include a sequence alignment (based on the structure) of the b3-b4 region in other PDZ domains (at least with the PDZ domains used in figure 5).

This is a valid point and we now include a structure-based sequence alignment of structurally related PDZ domains that were tested for OTULIN binding (**new Supplementary Figure 3c**). The alignment that the $\beta 3$ - $\beta 4$ loop is unique for the SNX27 PDZ domain. While other PDZ domains contain an insertion between this region, notably in SHANK 1-3 PDZs, the insertion does not adopt any known secondary structure and is unable to form any stable association with OTULIN as demonstrated by the ITC data.

4. Many experiments of the manuscript rely on mutagenesis of either OTULIN or SNX27 PDZ domain. Despite that ITC results are quite convincing and make sense with the whole story, it would be nice to check the quality of the point mutants. For example with OTULIN point mutants, since it is a deubiquitinase, it would be relatively easy to conduct some deubiquitinating activities comparing WT and mutants.

As suggested, we have tested cleavage activity of the most important SNX27-binding defective OTULIN mutants Δ ETSL, T350D and E85R/D87R in vitro and found no impairment in cleavage of Met1 chains (**new Supplementary Figure 2b**). In addition, we confirmed that OTULIN mutants Δ ETSL, T350D and E85R/D87R are still able to disassemble Met1-Ub chains generated by the LUBAC in cells (**new Supplementary Figure 2c**).

5. I think that Figure 4b should be better displayed in stereo.

We have now included a stereo-view of the second binding site to aid visual inspection of the site.

6. At the end of page 13, it is strange that SNX27 is not shifted to the VPS26-35 retromer fractions in OTULIN KO gel filtration analysis. Perhaps this is because of the weak interaction between SNX27 and VPS26-35 retromer ???. The authors should at least discuss this point.

Indeed, we have performed new experiments that clearly demonstrate a weak interaction of the purified SNX27 PDZ and VPS26A (please see below the reply to point 7). Based on the new results, we conclude that *'Most likely the low affinity of SNX27 and VPS26A explains why the retromer complex is not maintained during chromatography.'* (see **page 21**).

7. Since the authors claim that there is a competition in the binding between SNX27 and OTULIN or VPS26-35 retromer, in light of figure 5, it would be interesting to conduct some competition binding analysis in vitro. I think that this experiment might be quite informative to support the major hypothesis

of this work on the control of cargo binding by OTULIN. There should exist a competition between OTULIN and VPS26-35 to bind SNX27.

We tried to perform in vitro competition experiments, but we were unable to detect binding of GST-tagged SNX27 PDZ to VPS26A. Interestingly, despite the published structure of SNX27 PDZ in complex with VPS26A, no in vitro binding studies have been conducted (see Gallon et al., 111(35): E3604, 2014). Thus, we performed ITC with SNX27 PDZ and VPS26A and determined a K_D of 27 μ M (**new Figure 5e**), which is an almost 1000-fold lower affinity compared with the affinity of OTULIN and SNX27. We also performed size exclusion chromatography using purified proteins (**new Supplementary Fig. 4**). While OTULINcat and SNX27 PDZ formed a complex of 1:1 stoichiometry, SNX27 PDZ only slightly trailed towards VPS26A-containing fractions, validating the rather loose association of the two purified proteins, which most likely results from high K_{on}/K_{off} rates (see also point 6).

The data demonstrate that in vitro there is a strong balance for forming SNX27-OTULIN over SNX27-VPS26A complexes. However, in cells and in the context of a membrane-bound endosomal complex this balance may be shifted towards SNX27-retromer complexes. We modified the Discussion on **page 19 and 20**.

8. Fig.7c is quite relevant to show the decrease in the SNX27-cargo binding by OTULIN expression, it would be important to show some statistics based from the WB analysis.

We quantified binding of all detected cargos to GFP-SNX27 or endogenous SNX27 from four independent experiments. Quantifications of relative binding for each individual cargo are shown in the **new Supplementary Figures 7a, b, e**. For proper normalization of cargo binding, we used GFP-SNX27 (**S7a**) or SNX27 precipitation (**S7b, e**) and therefore we did not quantify loss of binding in SNX27 KO cells. From these data, we determined cargo binding for all detected cargos, adding up to a cumulative score for relative SNX27 cargo binding with statistics as shown at the bottom of the WB in **Figures 7d-f**.

9. In fig 7b I don't observe the GFP-Trap precipitated retromer subunit VPS26 and VPS35, as stated in page 15. Please confirm or correct.

VPS26 and VPS35 interaction is shown in the previous figures (now Figure 5f, g) and we have corrected this mistake.

Reviewer #2 (Remarks to the Author):

In the manuscript entitled 'Regulation of the endosomal SNX27-retromer by OTULIN', Weber and her colleagues have identified a new interacting partner of OTULIN by mass spectrometry. The authors used the biotin-labelled OTULIN activity-based probe (ABP), which made it success to identify new interacting proteins including SNX27. Interaction of endogenous proteins of OTULIN and SNX27 was confirmed in lysates derived from different cell types. The interaction is through the SNX27-PDZ and the c-terminal motif of OTULIN, mediated by the second interface giving specificity. Biologically, this interaction does

not affect OTULIN deubiquitinase activity and does not affect LUBAC-dependent cellular signalling. SNX27 is known to interact with VPS26A to form a retromer complex and interaction with OTULIN is mutually exclusive, thus antagonizing the endosomal SNX27-VPS26/VPS35 retromer complexes in cells. The model cargo glucose transporter 1 (GLUT1) loading to SNX27 PDZ was impaired in the presence of OTULIN. In this way, the authors demonstrated for the first time that OTULIN was shown to be involved in cell trafficking and endosome-to-plasma membrane recycling. The manuscript is very well written and the biochemical properties of OTULIN-SNX27 interaction are shown very clearly and in a convincing manner. Most importantly, this is totally a novel function of OTULIN. The biological implication in the regulation of endosomal SNX27 retromer by OTULIN is very interesting yet this part requires more investigation.

Major points:

1. Figure 7f. It is important to examine the GLUT1 surface expression level in different cell types not only in U2OS. Moreover, for making a conclusion of OTULIN functions in the regulation of the retromer complex, it is important to investigate different target surface proteins, at least one more, regulated by the OTULIN-SNX27 complex, possibly by using a similar approach as used in Fig 7f.

We have performed several experiments to strengthen the biological relevance of our finding as outlined in the introductory reply at the beginning of the response letter. This includes experiments in which we transduced GFP-SLC1A4 (an amino acid transporter) to validate the negative impact of OTULIN on SNX27 trafficking using a second cargo in U2OS and HeLa cells (**new Figures 8b, c, Supplementary Figure 8b, c**). Increased lysosomal localization of SLC1A4, which is caused by decreased surface expression is shown upon OTULIN expression. Despite the fact that surface proteomics defined many SNX27-dependent recycling cargos (Steinberg al., Nat Cell Biol 15: 461, 2014), there are hardly any assays for measuring recycling of individual cargos. Disrupted plasma membrane recycling for GFP-SLC1A4 in SNX27 KD cells has been demonstrated (Kvainickas et al., J Cell Sci 130: 382, 2017).

In addition, we strengthened our assumptions by showing effects on GLUT1 trafficking after siRNA KD of OTULIN, SNX27 or OTULIN/SNX27 (**new Figure 8d-i**; please see also preceding note).

Minor points:

1. Supp Fig 4. It is critical to examine TNF-induced phosphorylation and degradation of I κ B α in the OTULIN KO clones as a control.

We show in a **new Supplementary Figure S6c** that increased I κ B α phosphorylation and degradation after TNF α stimulation in OTULIN KO cells. We also added HOIP KO cells, which display reduced I κ B α phosphorylation and degradation in response to TNF α .

2. Having a schematic of the OTULIN function in the regulation of the retromer complex is suggested to help readers.

We agree and we provide a scheme for the dual function of OTULIN in the cleavage of Met-1 Ub chains conjugated by LUBAC and antagonizing cargo recycling by the SNX27-retromer in **Figure 8h**. The model is described at the beginning of the Discussion on **page 17**.

3. in page 5 in the last paragraph; 'To prove that SNX27 is directly associating with OTULIN in cells'.... The author's approach can show that they make a complex in cells but cannot show 'direct' association. Thus, it is suggested to rephrase this part.

We agree and rephrased this sentence saying now: '*To confirm the SNX27-OTULIN interaction,...*'.

Reviewer #3 (Remarks to the Author):

The authors have identified a PDZ-PDZbm molecular interaction between SNX27 and OTULIN which appears to be stabilized by a second interface. The biochemical/structural identification of this interaction is complete. However, the context of this interaction within the cell is unclear and needs to be further investigated. Most of the evidence presented is consistent with this interaction occurring in the cytoplasm rather than on membranes. This needs to be directly examined. The conclusions about the relevance of this molecular interaction to the functional action of SNX27 is weak and maybe indirect. A real option is that high levels of expression of OTULIN sequesters SNX27 through a molecular interaction in the cytoplasm. The evidence that this interaction is actually of physiological relevance is indirect and evidence supporting it needs to be improved.

Specific Points to be addressed

1) The molecular interaction work is messy in that authors nominate to use a range of engineered or endogenous OTULIN constructs across the different experiments. These experiments need to be repeated using a consistent bait for the different co-ip experiments.

We are surprised about this harsh assessment. We feel that it is a strength of our study that we identified and verified the OTULIN-SNX27 interaction using different cell lines and primary cells under varying experimental conditions.

Just to summarize the experimental flow: The initial identification and characterization of the OTULIN-SNX27 interaction was completely done using Jurkat T cells (Figure 1a-h). Identification by MS was done using the highly selective biotin-coupled OTULIN activity-based probe that quantitatively precipitates OTULIN and is therefore an excellent tool to detect new interactors (Figure 1a, b) (Weber et al., Cell Chem Biol. 2017). MS data were verified also taking advantage of CRISPR/Cas9 OTULIN and HOIP KO Jurkat T cells to prove the selectivity (Figure 1c-e). In reconstituted OTULIN KO Jurkat T cells we verify OTULIN-SNX27 interaction by directly precipitating OTULIN (Figure 1f, g). Finally, we demonstrate endogenous OTULIN-SNX27 interaction by IP in both directions using anti-OTULIN or anti-SNX27 antibodies (Figure 1h). So far, all experiments have been performed in Jurkat T cells and we next investigated that this is not a T cell-specific interaction by showing endogenous binding in HEK293 cells and murine embryonal fibroblasts (MEF) (Supplementary Fi. 1e and Figure 1i). We have tested more cell lines, but we have actually restricted ourselves, because we think the point is clear. Finally, we demonstrate that the OTULIN-SNX27 interaction is also seen in primary cells and for this we used murine CD4 T cells purified from the secondary lymphoid organs (spleen and lymph nodes) (Figure 1j).

We think that we provide a comprehensive description of the identification and verification of the OTULIN-SNX27 interaction. The other reviewers explicitly acknowledged the presentation, the experimental flow, the clarity, the technical quality and the convincing manner how the biochemical properties of the OTULIN-SNX27 interaction are shown.

2) Retromer will associate with SNX27 on endosome membranes in the presence of cargo. The current working model supported by the biochemistry is that retromer will not associate with SNX27 in the cytosol. As OTULIN-SNX27 interaction appears to be in the cytosol it will not be competing with OTULIN.

We are not sure about the point of the reviewer, but we assume that the last sentence is supposed to read '*As OTULIN-SNX27 interaction appears to be in the cytosol it will not be competing with retromer*' (instead of OTULIN). Indeed, this is one major point of our study and we added new data in **Figure 6** and **Figure 8** (see details below) to demonstrate that OTULIN-SNX27 complexes are localized in the cytosol and antagonize endosomal SNX27-retromer assembly and trafficking. As suggested by reviewer 2 we have added a scheme for the function of OTULIN (**new Figure 8h**). To make this clearer, we now explain at the beginning of the Discussion (**page 17**):

'By identifying SNX27 as an interactor of OTULIN, we have uncovered an unexpected role of OTULIN in the regulation of endosome-to-plasma membrane recycling (Fig. 8h). Contrary to the well-described function of OTULIN in counteracting Met1-linked ubiquitin chains synthesized by LUBAC, the impact of OTULIN on the SNX27-retromer is independent of DUB activity and is driven by the high affinity binding of OTULIN to the cargo binding pocket in the SNX27 PDZ domain. OTULIN-bound SNX27 is not found at early endosomes, but localizes to the cytosol. In line, OTULIN is not a trafficking cargo, but counteracts SNX27 recruitment to typical cargos and retromer subunits VPS26/VPS35 thereby preventing assembly of a functional SNX27-retromer complex. In consequence, OTULIN antagonizes SNX27-dependent recycling of the nutrient transporters GLUT1 and SLC1A4 and potentially many cell surface proteins.'

3) Direct evaluation of the impact OTULIN has on the membrane recruitment of SNX27 needs to be examined.

We repeated the experiments showing localization of SNX27 and OTULIN using now lentiviral expression in HEK293 cells (**new Figure 6b, c**). While GFP-SNX27 localizes to EEA1-positive endosomes, OTULIN and OTULIN C129A largely abolish the endosomal localization, leading to a more homogenous cytosolic distribution. However, SNX27 is retained on endosomes in cells co-expressing the OTULIN Δ ETSL mutant that lacks the PDZbm. Furthermore, we conducted new experiments showing that viral expression of OTULIN WT, but not Δ ETSL, removes endogenous SNX27 from EEA1 positive endosomes in HEK293 cells (**new Figure 6d**). Finally, SNX27 association with EEA1 positive endosomes is slightly enhanced in OTULIN KO HEK293 cells (**new Figure 6e**).

Despite its endosomal localization, we always see with GFP-SNX27 or endogenous SNX27 a staining outside the endosomes and vesicular compartments in the cytosol, suggesting that not all SNX27 is attached to the endosomes and the retromer.

4) Figure 6c magnification details required.

We added scale bars (10 μ m) to all panels with microscopy images.

5) Figure 6c data is poor and potentially selective. The level of expression GFP-SNX27 is lower in the endosome associated images compared to the others. More representative images need to be presented. In fact it would be best just to examine the impact of endogenous SNX27 in these experiments.

As outlined in the response to point 3, we have repeated the co-localization after lentiviral transduction of GFP-SNX27 and RFP-OTULIN (**new Figure 6b, c**). We show confocal images of a number of cells. Lentiviral constructs are expressed homogeneously in the cells excluding that effects are only caused by strong expression in some cells. In addition, endogenous SNX27 is less co-localizing with EEA1-endosomes in RFP-OTULIN versus RFP-OTULIN Δ ETSL expressing HEK293 cells (**new Figure 6d**) and SNX27-EEA1 co-localization is mildly enhanced in OTULIN KO HEK293 cells (**new Figure 6e**). Thus, OTULIN is able to drag SNX27 away from the early endosomes.

6) The deltaETSL appears to be within the nucleus of the cell shown.

The reviewer is right that it appears in the images that OTULIN Δ ETSL is slightly more in the nucleus, which we mention now on **page 12-13**. Slightly enhanced nuclear localization may be caused by loss of SNX27 binding, but we did not follow up on this, because it is not important in this context.

7) Do not switch cell lines between the co-ip (6a,b – HEK293) and IF 6c U206 cells. Repeat the experiments throughout this study within the same cell line – this will allow comparison of the results independent of the numerous differences between the cell types.

We performed all microscopy experiments in HEK293 cells (**Figure 6b-e**) to demonstrate localization.

8) “endosomal association of GFP-SNX27 was lost” is not supported by the data presented. You would not observe the endosome associated SNX27 in the images shown. Co-localisation studies need to be presented with endogenous SNX27 to make this conclusion. Inclusion of membrane fractionation would confirm these observations.

We rephrased this, because we agree that loss of SNX27 endosomal localization may be exaggerated and cannot be concluded from this type of experiment. We state now that ‘...*lentiviral co-expression of RFP-OTULIN diminished endosomal association of GFP-SNX27...*’. As outlined above, we demonstrate decreased association of endogenous SNX27 with EEA1-positive endosomes after viral OTULIN expression (**new Figure 6d**). We think that the new data and experiments strongly support the conclusion that endosomal association of SNX27 is altered by OTULIN binding and we do not think that membrane fractionations, which can often misleading due to problems with clean separation of the compartments, will add novel insights.

9) The level of over-expression of OTULIN needs to be documented. A trivial explanation of these results is that non-physiological levels through extreme over-expression may simply saturate the system and just sequester SNX27 (dominant interfering). This is fine to disrupt the system but is not good evidence that the molecular interaction is actually physiologically relevant.

Our assumptions are not solely based on overexpression of the proteins. The endogenous OTULIN-SNX27 complex was detected after size exclusion chromatography and complex formation was verified in KO cells (**Figure 6a**). Thus, in line with the high affinity in vitro, the cellular data confirm that the OTULIN-SNX27 complex is stable, at least more stable than the association of SNX27 with VPS26. We have no OTULIN antibody for indirect immunofluorescence, but we no longer use transfection which may result in very high expression in some cells, but we switched to milder and homogenous lentiviral expression (**new Figure 6b-c**). Also, OTULIN KO leads to mildly enhanced SNX27 localization at early endosomes (**new Figure 6e**). Further, we show enhanced cargo binding to SNX27 in OTULIN KO cells (**Figure 7e**). Most important, we conducted siRNA KD experiments to confirm that depletion of OTULIN influences SNX27-dependent GLUT1 trafficking (**new Figure 8d-i**). Taken together, these results exclude that the effects are merely caused by unphysiological overexpression.

10) Is the function of SNX27 impacted in the OTULIN KO cell lines you have generated?

We show in **Figure 7d and e** that SNX27 binding to cargos is impacted by loss of OTULIN, which is rescued by OTULIN WT but not Δ ETS1. Expression of many surface receptors or transporters is essential for cell viability and thus effects in KO cells may be influenced by clonal selection or adaptation. Therefore, we decided to address this question by siRNA KDs. As described in detail above, OTULIN depletion mildly increased GLUT1 at the plasma membrane, while the combined OTULIN/SNX27 KD restored GLUT1 surface expression compared to SNX27 single KD cells (**new Figure 8d-f**). In contrast, while siSNX27 increased GLUT1 on the lysosomes, siOTULIN alone did not affect lysosomal sorting of GLUT1. However, depletion of OTULIN in SNX27 KD cells prevented increased GLUT1 sorting to lysosomal vesicles caused by the SNX27 single KD (**new Figure 8g-i**). Thus, we provide evidence that endogenous OTULIN antagonizes SNX27-retromer dependent endosome-to-plasma membrane recycling.

11) The data in 6e was concluded that the “majority of OTULIN eluted together with SNX27”. However, SNX27 is predominantly on membrane associated on endosomes by IF shown. Is OTULIN also predominantly on endosomes. The inconsistency of these observations needs to be discussed.

We have no OTULIN antibody for IF, but lentiviral RFP-OTULIN is not on endosomes but evenly distributed in the cytosol (see **new Figure 7c, d**). IF demonstrates that SNX27 associates with endosomes, but we and other publications also detect cytosolic staining. Based on IF it is not possible to quantify how much SNX27 is on endosomes and how much is in the cytosol, especially since accumulation on specific membranes in clusters will be favored in IF-detection. For the gel filtration we discuss on **page 19-20**:

*‘Comparative analytical size exclusion chromatography from cytosolic extracts of Jurkat T cells revealed that endogenous OTULIN and SNX27 are found in a quite stable complex with an apparent 1:1 stoichiometry, which was also determined in vitro. The OTULIN-SNX27 complex is not bound to the VPS26-VPS35 retromer complex or LUBAC. Given the crucial function of SNX27 in retromer-dependent endosome-to-plasma membrane cargo trafficking, this stable association with OTULIN was unexpected. **Even though preparation of extracts for gel filtration may favor cytosolic components over proteins bound to endosomal membranes, the data suggest that only a fraction of SNX27 associates with the retromer.**’*

Overall, we do not think that there is an inconsistency, but the data reflect differences in experimental procedures that need to be considered. So, while IF may favor endosomal SNX27 detection, gel filtration may bias for detection of cytosolic SNX27 complexes and of course based on gel filtration we would never claim that SNX27 is not a part of the endosomal retromer complex.

We deleted terms *‘the majority of’* or *‘predominate’* association to avoid confusions. We hope that the reviewer acknowledges the striking data that OTULIN and SNX27 in vitro (**new Supplementary Figure 4**) and in cell extracts (**Figure 6a**) form a distinct and stable complex of 1:1 stoichiometry.

12) Western blotting/co-immunoprecipitation data should be quantified incorporating data from at least 3 independent experiments. This is highlighted in Figure 7 in which the level of co-ip between the otulin ko (d) and mock (e) which should be the same.

We quantified binding of all detected cargos to GFP-SNX27 or endogenous SNX27 from four independent experiments. Quantifications of relative binding for each individual cargo are shown in the **new Supplementary Figures 7a, b, e**. For proper normalization of cargo binding, we used GFP-SNX27 (**S7a**) or SNX27 precipitation (**S7b, e**) and therefore we did not quantify loss of binding in SNX27 KO cells. From these data, we determined cargo binding for all detected cargos, adding up to a cumulative score for relative SNX27 cargo binding with statistics as shown at the bottom of the WB in **Figures 7d-f**.

13) 7f – how was surface GFP differentiated from intracellular GFP by FACS?

Just like for FACS in immune cell phenotyping, staining was performed using GLUT1 binding reagent GLUT1.RBD.GFP (Metafora Biosystems) on intact cells and not on permeabilized cells. Permeabilization is required to stain for intracellular proteins. Furthermore, incubation times were short (20 min) excluding that significant amounts of GLUT1 from the surface and in complex with the GFP-labelled detection reagent can be internalized. By gating on living cells in FSC/SSC we exclude dead and disrupted cells from FACS analyses. Finally, by showing reduced GLUT1 cell surface expression after SNX27 KD, we verify that GLUT1 surface staining in this assay is depending on SNX27 and the essential retromer component ANKRD50 (**new Figure 8d-f; new Supplementary Figure 8d-f**). Overall, we think that the assay is an excellent tool, because it is able to quantify GLUT1 directly on the surface, instead of indirect correlations based on lysosomal mis-sorting.

14) A more detail investigation of the OTULIN KO cell line needs to be performed to determine if this impacts on SNX27 function. This should examine the relative distribution of SNX27 (does it impact on recruitment to retromer positive endosomes); SNX27 cargo (reduced retrieval to plasma membrane ;higher endosome associated pool of molecules – likely higher rate of degradation). These phenotypes are then rescued by expression of OTULIN (at endogenous levels) but not OTULIN deltaETSL.

We demonstrate that SNX27 localization on the endosome is augmented in OTULIN KO KEK293 cells (new Figure 6e). We show augmented binding of cargos to SNX27 in OTULIN KO cells (Figure 7e, f) and there is no indication for higher rate of degradation caused by the absence of OTULIN. Finally, as stated above, we used SNX27, OTULIN as well as SNX27/OTULIN KD to monitor the effects on GLUT1 trafficking (new Figure 8d-i). Together with the comprehensive structural and biochemical investigations, the data clearly demonstrate that OTULIN impacts on SNX27 function.

Reviewer #4 (Remarks to the Author):

In the present study, the authors identify a novel interaction between the ubiquitin modifying enzyme Otulin and the retromer component SNX27. Otulin is shown to bind to SNX27 in a non canonical way as it interacts in a bivalent way with this sorting nexin. The PDZ ligand engages the PDZ binding pocket while a second interface contacts SNX27 at a site that partially overlaps with the VPS26 binding site that connects SNX27 to retromer. Because of this mode of interaction, Otulin is demonstrated to have much higher affinity for SNX27 than “regular” PDZ ligands. Since Otulin can displace SNX27 from its cargoes, it is potentially a novel regulator of the SNX27-retromer.

Technically, the study is of very high quality. The novel interaction is clearly delineated and supported by extensive crystallography and convincing measurements of the respective affinities. Otulin is surely a high affinity interactor of SNX27. That said, some doubts remain about the functional part of this study and I am not completely convinced that Otulin is in fact a regulator of the retromer pathway. SNX27 interacts with a wide range of PDZ ligands, most of them located at the C-terminus of integral membrane proteins. There are, however, many intracellular, cytosolic proteins that were shown to engage the SNX27 PDZ domain such as DGKz, ZO2, ANKRD50 and many others. Steinberg et al. (2013, Nature Cell Biology, supplementary table 1) have performed quantitative proteomics on GFP-SNX27. While Otulin (FAM105b) was readily detected, it is just one of many PDZ bearing proteins and the proteomics data (score, PSMs) does not indicate a particularly high stoichiometry.

We think that the reviewer raises very interesting points that address unsolved issues in the field of SNX27 retromer research. Of course one important question is, why is the SNX27-retromer binding to cytosolic proteins that are apparently not recycled out to the plasma membrane? How are cargos selected for loading and how is unloading achieved? We will not be able to answer all these questions, but we introduce a new player to the field. However, we have performed a number of new experiments comprising cellular localization (new Figure 6b-e), GFP-SLC1A4 as a second SNX27 cargo (new Figure

8b,c, Supplementary Figure 8b, c) and KD of OTULIN (**new Figure 8d-i**), which all support our conclusion that OTULIN affects recycling via the SNX27 retromer pathway.

SNX27 was the outstanding interaction partner when pulling on OTULIN (**Figure 1f**), but OTULIN has also been identified as a SNX27 interaction partner (Steinberg et al., NCB 2013; Tello-Laffoz et al., Traffic, 2017). Given the high affinity, the reviewer is concerned that OTULIN should be a more outstanding interactor in such lists. However, of course to allow transport the system needs to be dynamic and flexible and we do not want to claim that cellular SNX27 is almost exclusively bound to OTULIN. We deleted terms '*the majority of*' or '*predominate*' association, which could be misleading. However, as a cargo-recruiter SNX27 is expected to bind to a large interaction network and several points need to be considered. We modified and extended the Discussion on **pages 20-21** to take up the concerns of the reviewer as follows:

'OTULIN has been identified as part of the SNX27 interactome, but despite the high affinity it was just one of many interactors^{18,33}. As a cargo-recruiter, SNX27 is expected to bind to a large interaction network and several mechanisms may account for the broad specificity of SNX27. We know affinities of PDZbm peptides, but native cargos may differ markedly as seen in the case of OTULIN. Further, affinities may change in the context of the endosomal retromer complex. While OTULIN cannot bind VPS26A-bound SNX27, affinities of GLUT1 and Kir3.3 PDZbms to SNX27 in complex with VPS26A are increased by an order of magnitude²², indicating that upon retromer assembly conformational changes may favor cargo recruitment. In addition, cargo-engagement or release may be modulated by endosomal retromer interactors like ANKRD50, which as a PDZbm-containing protein enhances SNX27-retromer recycling²⁹. Finally, PDZbm phosphorylation could act as a switching mechanism for PDZ domain interactions³⁴. Interestingly, replacing threonine 350 with a phospho-mimetic aspartate in OTULIN PDZbm (-2 position) reduces the KD of OTULIN towards SNX27 to the affinity of a prototypic cargo. Thus, phosphorylation in the PDZbm or modifications in the secondary interface of OTULIN may destabilize the OTULIN-SNX27 complex. Vice versa, phosphorylation of some cargo peptides increases their affinity for SNX27, which may further shift the balance from OTULIN to cargo binding¹⁷. Detailed analyses will be required to assess the impact of conformational changes and post-translational modifications in the regulation of retromer assembly/disassembly.'

Furthermore, Kvainickas et al. (2017, Journal of Cell Science) showed that overexpression of the PDZ ligand bearing SNX27 interactor ANKRD50 has exactly the same effect on SNX27 cargo as Otulin: The ANKRD50 tail displaces GLUT1 (and likely other cargo) just as Otulin does and removal of the PDZ ligand from ANKRD50 suppresses this effect. This indicates that the displacement of cargo through overexpression of SNX27 interactors may not be a unique feature of Otulin. One could argue that Otulin is simply a high affinity interactor for SNX27 of unknown function that can displace other PDZ ligand bearing proteins from SNX27.

We are not claiming that displacement of cargos is a unique effect of OTULIN, but indeed the effect of ANKRD50 on recycling is very distinct from what we report on OTULIN. In fact Kvainickas et al (JCS, 2017) report that ANKRD50 exerts the opposite function by facilitating GLUT1 and SLC1A4 recycling. They

write in the abstract: '*... ANKRD50 simultaneously engages multiple parts of the SNX27–retromer–WASH complex machinery in a direct and co-operative interaction network that is needed to efficiently recycle the nutrient transporters GLUT1 (also known as SLC2A1) and SLC1A4.*'

They demonstrate cooperative binding of ANKRD50 to multiple surfaces in the SNX27-retromer-WASH complex to augment recycling. In contrast, we show that OTULIN binds selectively and with high affinity the SNX27 PDZ domain and thereby impairs cargo binding, retromer assembly and recycling. The interaction of ANKRD50 and the SNX27 PDZ domain is most puzzling as they write in the Discussion: '*In this context, the association of the ANKRD50 C-terminal PDZ-binding motif with SNX27 is the most puzzling interaction, as ANKRD50 would block the PDZ-binding pocket of this sorting nexin for SNX27 cargo like GLUT1 or SLC1A4 if it was constitutively bound to it. One possible function ... ANKRD50 displaces SNX27-bound cargo to release the cargo into recycling tubules for transport.*'

The mechanism is conceivable, but certainly not proven. Anyway, ANKRD50 augments and OTULIN antagonizes recycling, reflecting that multiple components must work together to allow faithful endosome-to-plasma membrane trafficking.

To address these concerns, the authors could demonstrate that the effect of Otulin knockout on SNX27 cargo binding is unique. For that they could use CRISPR or siRNA pools (Dharmacon smartpools, for example) to knockout/knockdown proteins such as ANKRD50, ZO2, DGKzeta, ABRO1 and analyze SNX27 cargo binding. Similarly, it would be nice if some of these proteins could be overexpressed to check whether this results in a similar loss of SNX27-cargo interaction as overexpression of Otulin. If they did, this would argue that overexpression of SNX27-PDZ interactors generally perturbs cargo binding. Clearly, the knockdown/knockout data is more relevant here, as the overexpression results may be meaningless.

The only feasible experiment at this stage is knock-down comparisons, but for these IPs we need a lot of cells making the experiment very challenging. Further, we doubt that such analyses can provide evidence for a unique role of OTULIN. We have no data on the relative expression of these PDZbm-containing proteins. How should we compare the effect of an OTULIN KD for instance to a DGK ζ KD? More important, what would the result for siDGK ζ (or other PDZbm cargos) imply for the regulation of SNX27 by OTULIN? We lack structural and biophysical insights how these interactors engage with SNX27 and potentially other PDZ-containing proteins or the retromer. So, if DGK ζ KD positively or negatively affects cargo loading, would that mean that the OTULIN-SNX27 interaction is less relevant?

Nevertheless, to take up the concerns of the reviewer, we used Dharmacon smartpool siRNAs to KD ANKRD50 and DGK ζ , for which we could get antibodies to verify the depletion in U2OS cells, and determined GLUT1 expression on the cell surface (**new Supplementary Figure 8d-f**). In line with its published positive function for SNX27-dependent GLUT1 recycling (Kvainickas, JCS, 2017), KD of ANKRD50 mildly decreased GLUT1 surface expression. To our knowledge, there are no data on the regulation of recycling by ZO2, ABRO or DGK ζ , but at least for DGK ζ we did not observe a significant effect on GLUT1 surface abundance. Thus, the data show that not all PDZbm-containing proteins are impairing SNX27-retromer recycling like OTULIN. We do not want to claim that the data demonstrate

that the function of OTULIN on SNX27 cargo binding and recycling is unique, because we feel for the reasons listed above this would be an over-interpretation. Certainly, how cargo selection is controlled, how various PDZbm-containing proteins may cooperate in such selection and if there may be functions beyond retromer regulation are highly interesting points, but it is beyond the scope of our study, which defines OTULIN as a new player in the SNX27 network.

Additional points:

The authors claim that Otulin displaces SNX27 from endosomes, probably by outcompeting binding to VPS26. They also state that only a fraction of SNX27 is retromer associated, while a significant fraction of SNX27 co-migrates with Otulin. In my experience working with SNX27, both in fixed cells and in living cells, SNX27 is very crisply endosomal with minimal cytosolic background signal. This is odd if Otulin really keeps the majority of SNX27 in a cytosolic complex. Does SNX27 recruit Otulin onto endosomes?

We have no OTULIN antibody for IF, but lentiviral RFP OTULIN is not on endosomes but evenly distributed mostly in the cytosol (see **new Figure 6c, d**). IF demonstrates that SNX27 associates with endosomes, but here and in other publications cytosolic staining is also detected for SNX27. In our view, it is not possible to quantify how much SNX27 is on endosomes and how much is in the cytosol based on IF, especially since accumulation on specific membranes in clusters may be favored in IF-detection. In contrast, gel filtration may favor cytosolic complexes (see response to point 11 of reviewer 3) and we propose that two pools of SNX27 exist, one at the endosome in complex with the retromer and one in the cytosol in complex with OTULIN. A scheme in **Figure 8h** illustrates the model and we added a description on **page 17**.

Figure 6C: The colocalization data is not very convincing. The authors should moderately overexpress SNX27 and Otulin and analyze the subcellular distribution of SNX27 using a good confocal microscope and high zoom. GFP-SNX27 tends to be quite cytosolic when overexpressed too much so this needs to be done carefully. Ideally, low level lentiviral expression of SNX27 is used as this results in SNX27 that is properly localized to endosomes.

We have repeated these studies using lentiviral expression of GFP-SNX27 and RFP-OTULIN in HEK293 cells (**new Figure 6b, c**). We obtain homogenous expression and show more cells and improved on image quality. We also include data showing re-localization of endogenous SNX27 from the endosomes after transduction of RFP-OTULIN WT, but not RFP-OTULIN Δ ETSL (**new Figure 6d**). Finally, SNX27 association with EEA1 positive endosomes is slightly enhanced in OTULIN KO HEK293 cells (**new Figure 6e**). In line, KD of OTULIN enhances GLUT1 expression on the cell surface (**new Figure 8d, e**). Taken together, these results suggest that OTULIN via binding to SNX27 can modulate accessibility of SNX27 for endosomal retromer trafficking.

Figure 7B: The authors state that GFP-SNX27 precipitates VPS26 and VPS35. While this is accurate, it is not shown in the panel.

We corrected this mistake. Precipitation of GFP-SNX27 with the retromer VPS26 and VPS35 is shown in Figure 5f and g and mentioned earlier.

Figure 7D: The authors need to show the levels of the respective cargoes in the Otulin KO cells. As it is, it is unclear whether the increase in SNX27 cargo binding is really caused by increased binding or whether there is simply more of the respective cargoes in the KO clone.

We included Western Blot to show the levels of respective cargoes in the lysates of WT and KO cells prior to IP now in **Figure 7e and f**. Clearly, differences in binding are not caused by changes in the expression of the cargoes. Of note, while KD of SNX27 was shown to cause a decrease in expression of many cargoes (e.g. see Steinberg et al., NCB 2013), we did not observe significant downregulation of cargoes in SNX27 KO cells. We think that this is due to clonal selection and adaptation after SNX27 KO in the cells and thus we switched to KD for the functional assays (see **Figure 8**)

GLUT1 data: The authors show that overexpression of Otulin reduces surface expression of GLUT1. It would be great if the authors knocked down Otulin and analyzed GLUT1 surface levels following the knockdown. If surface levels were increased, this would show that loss of Otulin promotes the activity of SNX27. The same could be done for SLC1A4, which is another SNX27 dependent surface channel (Kvainickas et al., 2017, Journal of Cell Science).

As mentioned earlier, we have performed siRNA KD of SNX27, OTULIN or combined SNX27/OTULIN KD and measured either GLUT1 surface expression (**new Figure 8d-f**) or determined GLUT1/LAMP1 co-localization (**new Figure 8g-i**). We measure increased GLUT1 surface expression after OTULIN KD. Moreover, combined OTULIN/SNX27 KD restored GLUT1 surface expression and impaired GLUT1 localization to lysosomes when compared to SNX27 KD alone. The results provide evidence that SNX27 and OTULIN participate in the same pathway to regulate endosome-to-plasma membrane trafficking. In addition, we provide data that OTULIN but not OTULIN Δ ETSL expression affects trafficking of GFP-SLC1A4 in U2OS and HeLa cells (**new Figure 8b, c, Supplementary Figure 8b, c**).

Minor points:

Page 5: The authors state that they tested whether OTULIN directly binds to SNX27. For that they perform the biotin probe IP in OTULIN KO cells, in which no SNX27 was precipitated. While this is a nice experiment, it does not prove that SNX27 directly interacts with OTULIN. This should be rephrased. Clearly, OTULIN is required to pull down SNX27 with their probe but this is not real evidence for direct binding as there may still be a third bridging molecule. The same also applies to the 2xStrep-Flag precipitation. The authors would need to purify the PDZ domain and OTULIN in bacteria and test for direct interaction to really prove this. Given that they have a crystal structure of the complex, this seems rather pointless so the authors could simply rephrase these sentences.

We agree and rephrased this sentence saying now: *'To confirm the SNX27-OTULIN interaction,...'*

We would like to point out that besides the crystal structure of the complex we demonstrate direct interaction in vitro by ITC. In addition, we now include size exclusion chromatography using recombinant OTULINcat and SNX27PDZ to show that both proteins form a complex of 1:1 stoichiometry in vitro (**new Supplementary Figure 4**).

REVIEWERS' COMMENTS:

Reviewer #1 (Remarks to the Author):

I think that the authors have addressed most of the points raised by all four reviewers by performing new experiments and new data analyses. The biological relevance of this novel regulation in cargo transfer is better sustained in the revised form of the manuscript.

Minor comment: Are you sure that the stereo figure 4b is OK ?. It looks flipped to me.

Reviewer #4 (Remarks to the Author):

The authors have added important data that make the study much more convincing. My doubts about the regulatory role of Otulin in the SNX27-retromer pathway have been resolved. I now support publication of this high quality study.

Reviewer #1 (Remarks to the Author):

I think that the authors have addressed most of the points raised by all four reviewers by performing new experiments and new data analyses. The biological relevance of this novel regulation in cargo transfer is better sustained in the revised form of the manuscript.

Minor comment: Are you sure that the stereo figure 4b is OK? It looks flipped to me.

We have investigated the panel and three persons were able to see the stereo view in the TIFF-file. Maybe it was slightly blurry due to the PDF conversion. We have slightly enlarged the panel.

Reviewer #4 (Remarks to the Author):

The authors have added important data that make the study much more convincing. My doubts about the regulatory role of Otulin in the SNX27-retromer pathway have been resolved. I now support publication of this high quality study.

We thank the reviewers and are very happy about the positive assessment and recommendation for publication.